# Widespread protein lysine acetylation in gut microbiome and its alterations in patients with Crohn's disease

Xu Zhang [1,2], Zhibin Ning[1,2], Janice Mayne[1,2], Yidai Yang[1,2], Shelley A. Deeke[1,2], Krystal Walker[1,2], Charles L. Farnsworth[3], Matthew P. Stokes[3], Jean-François Couture[1,2], David Mack[4], Alain Stintzi [1,2 ✉] & Daniel Figeys [1,2 ✉]

Lysine acetylation (Kac), an abundant post-translational modification (PTM) in prokaryotes, regulates various microbial metabolic pathways. However, no studies have examined protein Kac at the microbiome level, and it remains unknown whether Kac level is altered in patient microbiomes. Herein, we use a peptide immuno-affinity enrichment strategy coupled with mass spectrometry to characterize protein Kac in the microbiome, which successfully identifies 35,200 Kac peptides from microbial or human proteins in gut microbiome samples. We demonstrate that Kac is widely distributed in gut microbial metabolic pathways, including anaerobic fermentation to generate short-chain fatty acids. Applying to the analyses of microbiomes of patients with Crohn's disease identifies 52 host and 136 microbial protein Kac sites that are differentially abundant in disease versus controls. This microbiome-wide acetylomic approach aids in advancing functional microbiome research.

[1] Shanghai Institute of Materia Medica-University of Ottawa Joint Research Center in Systems and Personalized Pharmacology, University of Ottawa, Ottawa, ON K1H 8M5, Canada. [2] Ottawa Institute of Systems Biology and Department of Biochemistry, Microbiology and Immunology, Faculty of Medicine, University of Ottawa, Ottawa, ON K1H 8M5, Canada. [3] Cell Signaling Technology Inc., Danvers, MA 01923, USA. [4] Department of Pediatrics, Faculty of Medicine, University of Ottawa and Children's Hospital of Eastern Ontario Inflammatory Bowel Disease Centre and Research Institute, Ottawa, ON K1H 8L1, Canada. ✉email: astintzi@uottawa.ca; dfigeys@uottawa.ca

The intestinal microbiome is emerging as an important organ within the human body that actively interacts with its host to influence human health[1]. Dysbiosis of the intestinal microbiota has been reported to be associated with a myriad of diseases, including obesity, diabetes, Crohn's disease (CD), cancer, and cardiovascular diseases[2]. In the past few years, meta-omic approaches, including metagenomics, metatranscriptomics, and metaproteomics, have been applied to study the alterations of the microbiome composition and functions in patients with these diseases[3–6]. However, very little is known of the regulatory processes in the microbiome, such as post-translational modifications (PTMs) that are known to regulate the activity of proteins. In fact, there are currently neither published studies on the global and deep characterization of PTMs in the human microbiome nor published techniques for efficient PTM profiling at the metaproteome level.

Acetylation is an important PTM in both Eukaryotes and Prokaryotes[7,8]. In particular, lysine ($N_\varepsilon$) acetylation (Kac) has been shown to be involved in the regulation of various biological processes, including transcription and metabolism[8,9]. Compared with other PTMs that are commonly implicated in regulation of metabolic processes, such as phosphorylation, acetylation demonstrated higher levels in microorganisms[10]. In bacteria, up to 40% of proteins can be acetylated[11], due to the presence of both enzymatic and nonenzymatic acetylation mechanisms[12–15]. Protein Kac has been characterized in several single bacterial species, including *Escherichia coli*[13,16–18], *Bacillus subtilis*[19], *Salmonella enterica*[8], and *Mycobacterium tuberculosis*[20], and widely implicated in various microbial processes, including chemotaxis[21], nutrient metabolism[18], stress response[18], and virulence[22]. In *E. coli*, the enzymatic activities in acetate metabolism were regulated by acetylation[18]. On the other hand, metabolic intermediates of acetate metabolism, such as acetylphosphate and acetyl-CoA, can non-enzymatically acetylate metabolic enzymes or provide acetyl donor for enzymatic lysine acetylation. Therefore, microorganisms may evolve elegant mechanisms in regulating cellular metabolism through acetylation[12].

One of the most important metabolic functions of the gut microbiome is fermentation of indigestible dietary fibers to generate short-chain fatty acids (SCFAs)[23]. SCFAs can nourish the intestinal cells, maintain the acidic intestinal environment, and thereby protecting the intestinal barrier function[24]. Accumulating evidence suggests that intestinal SCFAs and SCFA-producing bacteria at least partially mediate the complex host–microbiome interactions that underlie the development of many diseases, such as CD[3,25]. Given the potential role of Kac in regulating SCFA metabolism[18,26], the study of Kac in human gut microbiome may aid in better understanding the role of gut microbiome in CD.

In this study, we first establish experimental and bioinformatic workflows for characterization of microbiome Kac. Briefly, an immuno-affinity-based approach is used for the enrichment of Kac peptides from the microbiome protein digests; the eluted peptides are analyzed with Orbitrap-based mass spectrometer (MS); and the MS data are then processed using an integrated metaproteomics/lysine acetylomics bioinformatic workflow that is developed in this study. In total, 35,200 Kac peptides corresponding to 31,821 Kac sites are identified from either human or microbial proteins. This study is a global characterization of Kac proteins in human microbiomes and achieves the highest number of site identifications in lysine acetylomic studies. We further apply the approach to study alterations of Kac in intestinal microbiomes of children with new-onset CD, which demonstrates the upregulation of Kac in host proteins, such as immune-related proteins, and downregulation of Kac in microbial proteins from the Firmicutes species that are known SCFA producers. This study provides an efficient workflow for studying lysine acetylome in the microbiomes, and our results provide additional information on the intestinal dysbiosis in pediatric CD.

## Results

**Integrated gut metaproteomic/lysine acetylomic workflow.** In this study, the proteolytic peptides generated from each microbiome sample were aliquoted for both metaproteomics and lysine acetylomics analysis. Kac peptides from the first aliquot were enriched using a seven-plex anti-Kac peptide antibody cocktail[27]; the second aliquot was directly analyzed for metaproteome profiling (Fig. 1a). An integrative metaproteomics/lysine acetylomics data-processing workflow was then developed based on our previously established MetaPro-IQ workflow[28] and MetaLab software tool[29] (Fig. 1b). We, and others, have previously shown that the Integrated Gene Catalog (IGC) database[30] performed similarly to the matched metagenome database for metaproteomic identification[31,32]. Therefore, in this study, we used the IGC database for the identification of both metaproteomic and lysine acetylomic data sets. Briefly, each of the raw files was first searched against the IGC protein database using MetaLab[29]; the parameters were as default, except that lysine acetylation ($m/z$ 42.010565, H[2]C[2]O) was added as an additional variable modification. The sample-specific databases for both aliquots of all samples were then combined and concatenated with a human protein database for peptide/protein identification and quantification of both data sets.

In total, this study identified 46,927 non-Kac peptides and 117 Kac peptides (171 Kac sites) from the metaproteomic aliquot; in contrast, 35,200 Kac peptides (31,821 Kac sites) and 7387 non-Kac peptides were identified from the lysine acetylomic aliquot (Fig. 1c; Supplementary Data 1). This result indicates a high efficient enrichment of Kac peptides with the anti-Kac antibody cocktail, namely 83% of the identified peptides were Kac peptides. Among the 31,821 Kac sites identified, 1662 sites were quantified in all six samples and 6206 in at least four samples (Supplementary Fig. 1). Among the 117 Kac peptides identified in the metaproteomic aliquot, only 6 were also identified in lysine acetylomic aliquot. Evaluating the overlap of identified Kac proteins with proteins identified in unenriched samples, this study identified 25,144 protein groups, and 3814 (15%) were only inferred from Kac modified peptides (Fig. 1d), suggesting an efficient enrichment of low abundant Kac proteins/peptides using the current enrichment approach.

**Characterization of the gut microbial Kac motifs.** Among the 31,821 Kac sites identified in lysine acetylomic aliquot, 31,307 were from microbes and 497 were of human origin (Fig. 1e). We first characterized the amino acid distribution surrounding the acetylated lysine, for both human and microbial Kac sites (Supplementary Fig. 2a), which suggests that both human and microbial Kac sites showed high frequency of glutamic acid (E) at −1 and leucine (L) and +1 positions. The microbiome protein Kac sites were frequently flanked by repeats of the small, hydrophobic amino acid, alanine (A); this was less common for the human protein Kac sites (Supplementary Fig. 2a). We further analyzed and visualized Kac protein motifs using pLogo[33] in a sequence window of 13 amino acids. For human Kac sites, significant over-representation was only observed for tyrosine (Y) at positions −4 and +1 (Supplementary Fig. 2b and Supplementary Data 2). In contrast, 68 significantly over-represented events were observed for microbiome protein Kac sites, with E and aspartic acid (D) at position −1 and phenylalanine (F) at position +1 being the most significantly overrepresented (Fig. 1f; Supplementary Data 2). Interestingly, small amino acid, alanine, was observed as significantly overrepresented at all the 12 positions assessed, and its

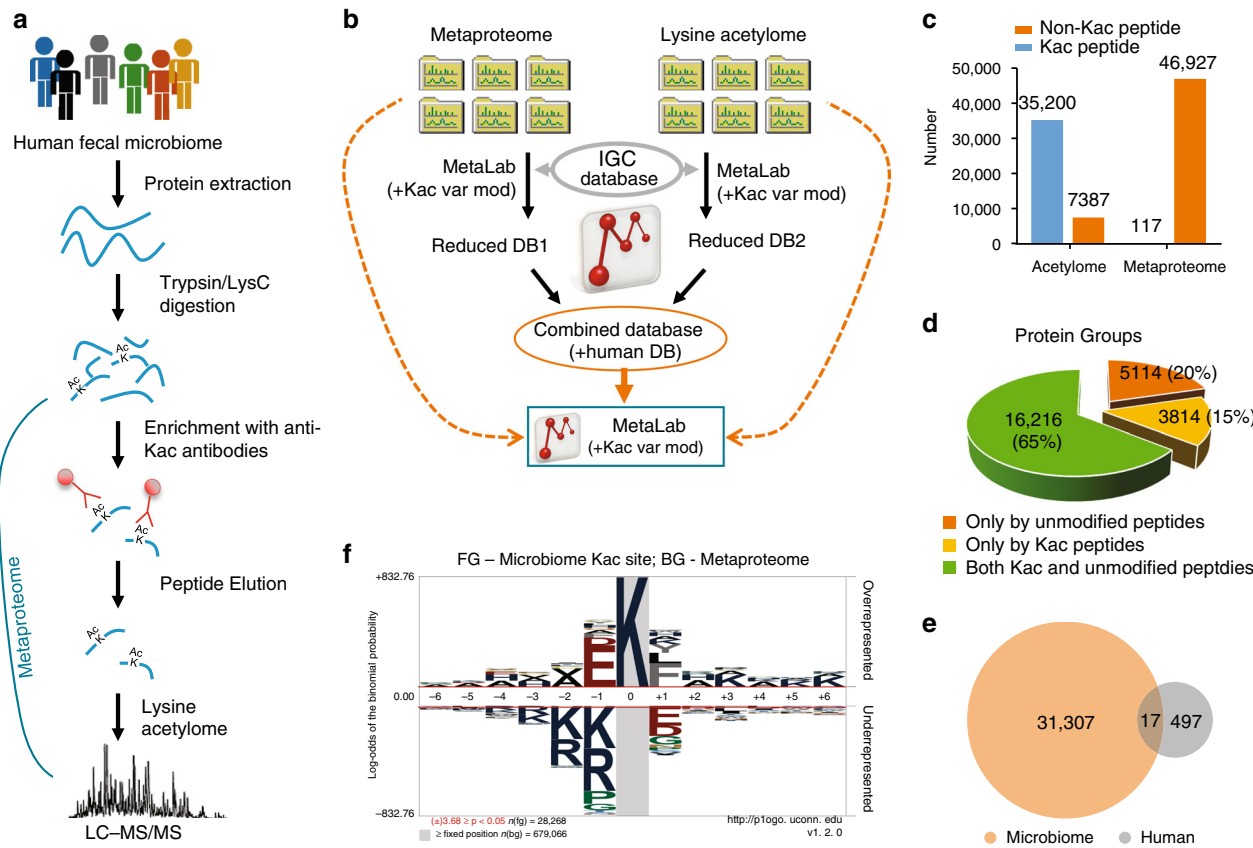

**Fig. 1 Experimental and bioinformatic workflows. a** Experimental workflow. **b** Integrated metaproteomics/acetylomics data-processing workflow. **c** Total number of identified Kac and non-Kac peptides in metaproteomic and lysine acetylomic aliquots, respectively. **d** Identified protein groups with non-Kac peptide and Kac peptide sequences in the whole data set (both lysine acetylomic and metaproteomic aliquots). **e** Venn diagram shows the overlap of identified human and microbial protein Kac sites. **f** pLogo of all identified microbiome Kac sites. The n(fg) and n(bg) values indicate the number of foreground and background sequences, respectively. The red horizontal bars on the pLogo correspond to a threshold of $P < 0.05$. Statistical significance of motif residues at given positions was assessed using binomial probability test. Source data are provided as a Source Data file.

occupancy frequency was >9% ($10.3 \pm 0.7\%$; mean ± SD) for all positions (Supplementary Data 2). In contrast, the median occupation frequency of all other significantly overrepresented events was 5.8% ($5.8 \pm 3.6\%$; mean ± SD). These observations further confirm the high frequency of alanine as well as acidic amino acids (E and D) near the Kac sites in microbiome proteins.

We then examined whether different bacteria showed different protein Kac motifs in microbiome. Among the five bacterial phyla (Firmicutes, Bacteroidetes, Actibobacteria, Proteobacteria, and Fusobacteria) with >100 Kac sites identified in this data set, high similarity between different phyla was observed (all showed high frequency of E at position −1 and F at position +1; Supplementary Fig. 3a–e). Firmicutes and Bacteroidetes are the two most abundant bacterial phyla in human gut microbiota and obtained the highest number of Kac sites identified. Motif analysis using pLogo identified 57 significantly overrepresented events (position—amino acid pairs) for Firmicutes and 41 significantly overrepresented events for Bacteroidetes. Interestingly, 35 out of the 41 significantly overrepresented events in Bacteroidetes were also significantly overrepresented events in Firmicutes (Supplementary Fig. 3f), indicating high conservation of Kac motif among different human gut microbial species.

**Phylogenetic variations of protein Kac level in microbiome.** Biodiversity analysis of the identified Kac peptides using Uni-pept[34] revealed that 28,321 peptides (80%) were assigned to the kingdom Bacteria and 24,785 peptides could be classified at the phylum level (15,170 from Firmicutes, 7876 from Bacteroidetes, and 1739 from other phyla; Fig. 2a; Supplementary Data 3). A high proportion of the Kac peptides was from bacteria belonging to four genera: *Prevotella*, *Faecalibacterium*, *Bacteroides*, and *Eubacterium* (Supplementary Data 3). To explore whether Kac levels differed by taxa, the ratio of relative abundance in lysine acetylome to that in metaproteome was calculated for each taxon (Fig. 2b). The genus *Fusicatenibacter* (mainly species *F. saccharivorans*) had the highest acetylome-to-metaproteome ratio (a median of 14.24), while *Homo sapiens* (human) had the lowest ratio (a median of 0.07) (Fig. 2b). This indicates that the protein acetylation level is much higher in Prokaryotes than human proteins in the microbiome samples. Firmicutes was the only phylum that showed significantly higher percentage in lysine acetylomic aliquot than that in metaproteomic aliquot ($P = 0.03$, paired Wilcoxon signed-rank test), while Actinobacteria and Proteobacteria showed significantly lower percentages in acetylomic aliquot (Fig. 2b). No significant difference was observed for Bacteroidetes, despite its lower acetylome-to-metaproteome ratio (Fig. 2b). We calculated the Firmicutes-to-Bacteroidetes (F/B) ratios based on the intensities of their distinctive peptides yielding an average of 6.35 in lysine acetylome, which was significantly higher than that of metaproteome (an average of 4.90; $P = 0.04$, paired Wilcoxon signed-rank test), further indicating higher protein acetylation levels in Firmicutes.

Acetylphosphate, a metabolic intermediate of SCFA production, has been shown to be a critical contributor for protein lysine acetylation in prokaryotes[13]. To study the association of

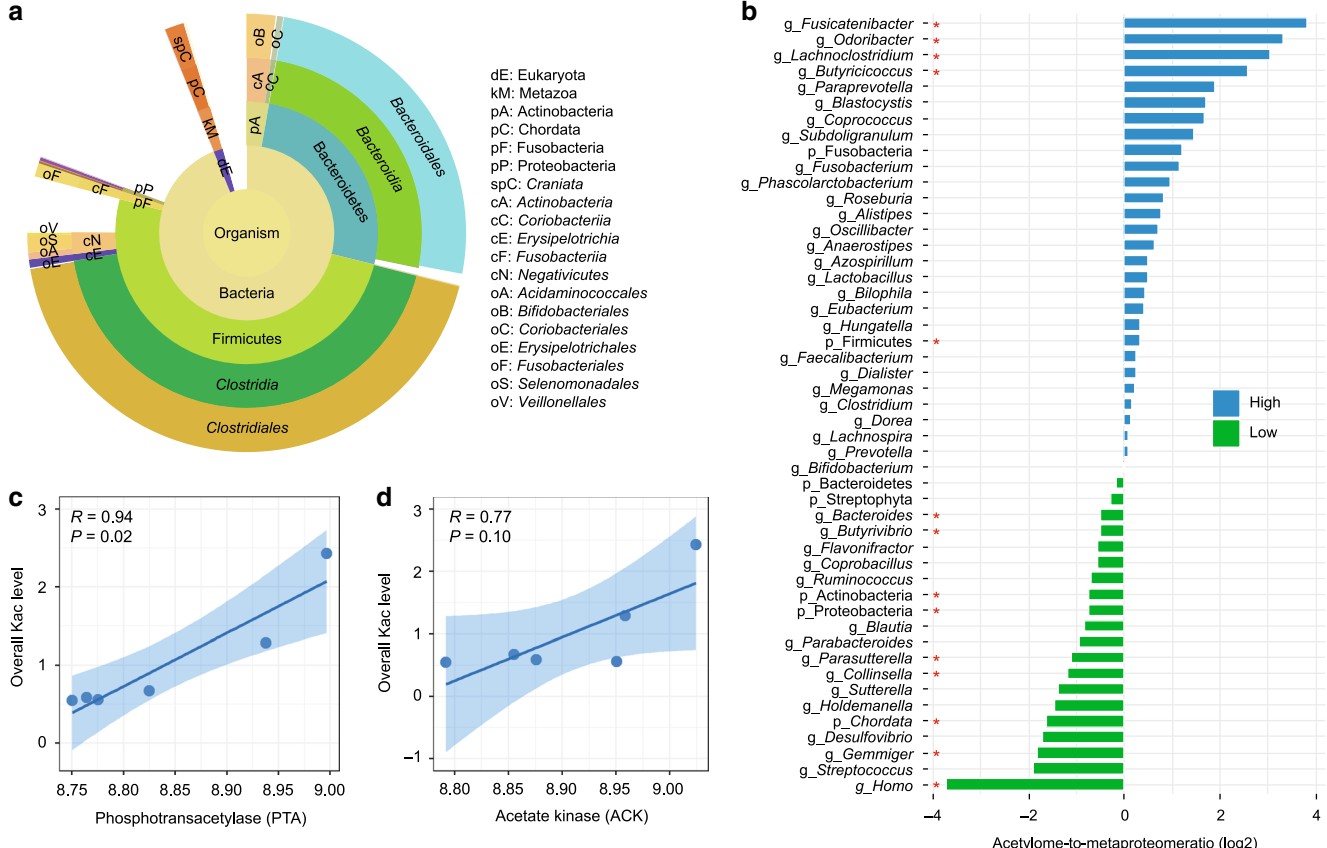

**Fig. 2 Taxon-specific lysine acetylation in human gut microbiome. a** Sunburst plot of microbial taxa that were assigned using all identified Kac peptides. Sunburst plot is generated using Unipept (https://unipept.ugent.be/). **b** Lysine acetylome-to-metaproteome ratios of quantified phyla and genera in human gut microbiome. The ratios were log2-transformed for plotting. High indicates higher lysine acetylation levels, and low indicates lower lysine acetylation levels. Red star indicates statistically significance ($P < 0.05$, paired, two-sided Wilcoxon signed-rank test) when comparing the percentage in lysine acetylomic aliquot with that in metaproteomic aliquot. **c, d** Correlations of overall Kac peptide abundances with the relative abundances of protein phosphotransacetylase (**c**) and acetate kinase (**d**) in metaproteome. Mean and 95% confidence interval of the correlation coefficient are shown as line and error band, respectively. Spearman's correlation $R$ and $P$ values are indicated. Source data are provided as a Source Data file.

acetylphosphate with global Kac levels in the microbiome, we correlated the abundances of acetylphosphate-producing enzymes, namely acetate kinase (ACK) and phosphate acetyltransferase (PTA), in the metaproteomic aliquots with the total abundance of all Kac peptides identified in acetylomic aliquots. Significant correlations were obtained for both PTA ($R = 0.94$; $P = 0.02$; Fig. 2c) and ACK ($R = 0.77$; $P = 0.10$; Fig. 2d). No or negative correlation was observed when correlating acetyl-CoA synthase (ACS) and acetyl-CoA acetyltransferase (ACAT) with total abundance of Kac peptides (Supplementary Fig. 4). Taxonomic assignment of the 87 proteins annotated as ACK and PTA identified in this study showed that 62 proteins belonged to Firmicutes, and 25 belonged to other phyla including Bacteroidetes. *Bacteroides* (14 proteins), *Blautia* (13 proteins), *Faecalibacterium* (11 proteins), *Lachnospira* (8 proteins), *[Eubacterium] rectale* (6 proteins), butyrate-producing bacterium SS3/4 (5 proteins), and *Prevotella* (5 proteins) were the genera/species with the highest number of ACK or PTA proteins identified in the metaproteomic aliquots (Supplementary Table 1). These results are in agreement with the taxonomic distribution of Kac peptides identified in lysine acetylomic aliquots (Fig. 2a), and again suggest that Firmicutes had higher protein acetylation levels in microbiome.

**Widespread protein Kac in gut microbial metabolic pathways**. To study the functional distribution of identified microbial Kac

proteins, we performed gene ontology (GO) term annotation of all identified Kac peptides, which showed that the top GO biological processes were translation (2642 peptides) and carbohydrate metabolism (1660 peptides) (Supplementary Table 2). Enzyme Commission (EC) number annotation yielded a total of 1025 enzymes from five enzyme classes (Supplementary Data 4). The most Kac modified enzymes identified in this study belonged to class EC 2 (transferases; mainly EC 2.7: transferring phosphorus-containing groups) and EC 1 (oxidoreductases). Functional enrichment analysis using the Clusters of Orthologous Groups (COG) database revealed that microbial Kac proteins were significantly enriched in energy production and conversion, transport, or metabolism of amino acid, nucleotide, lipid, and coenzyme, cell wall biogenesis, replication as well as secondary metabolite metabolism (Fig. 3a).

Kyoto Encyclopedia of Genes and Genomes (KEGG) annotation showed that 11,536 out of the 15,053 Kac proteins (76.6%) were mapped to 1354 KEGG orthologies (KOs) and 224 pathways. Among the 1354 KOs, 994 were enzymes and 626 were mapped to metabolic pathways. Fifty-one complete metabolic modules were constructed using Kac proteins, including glycolysis, citrate cycle, gluconeogenesis, pyruvate oxidation, and dissimilatory sulfate reduction (Supplementary Table 3). Intestinal microbiota is known to process complex carbohydrates, such as indigestible dietary fiber, to generate SCFAs that maintain the homeostasis of the intestinal microenvironment[23]. We found that

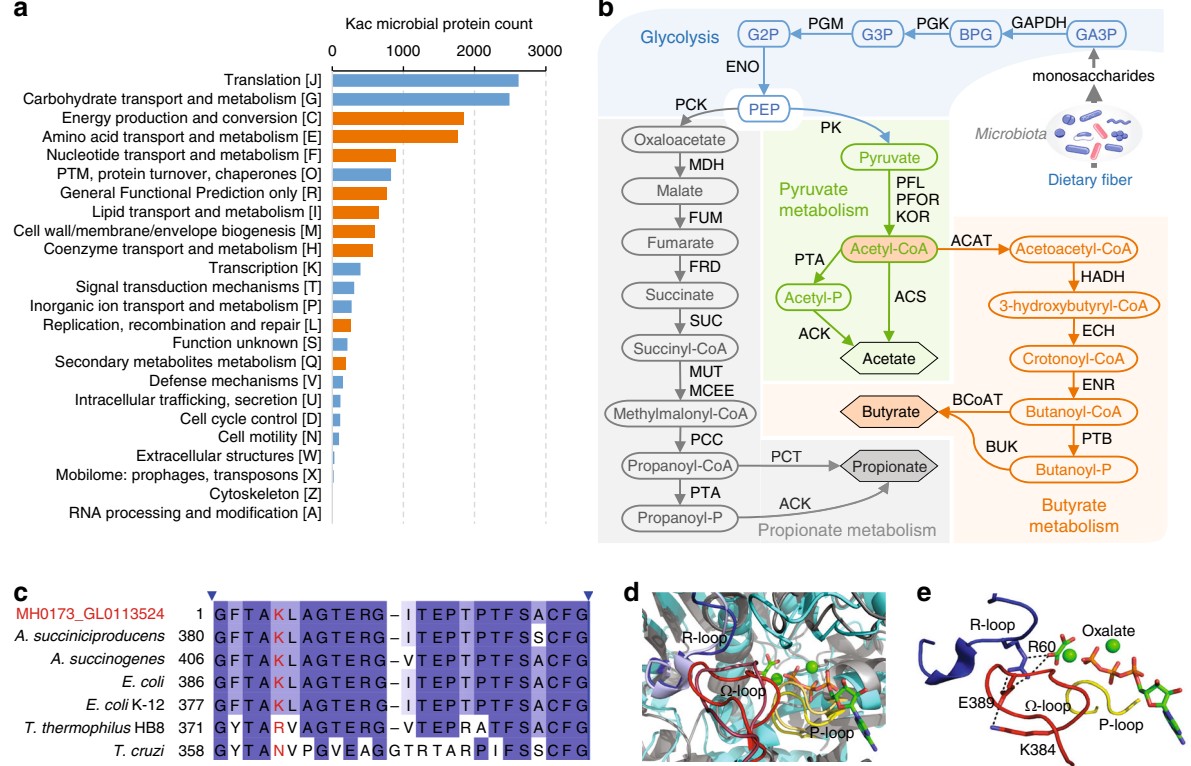

**Fig. 3 Functional characterization of identified Kac proteins. a** COG category distribution of microbial Kac proteins. Significantly enriched categories are highlighted in orange. Significance was determined with a hypergeometric test using the unmodified microbial proteins identified in the metaproteomic samples as background. Source data are provided as a Source Data file. **b** SCFA-producing metabolic pathways constructed using the identified Kac proteins. Identified Kac enzymes and metabolites are indicated using abbreviations as follows: GA3P glyceraldehyde-3-phosphate, BPG 1,3-bisphospho-glycerate, G3P glycerate 3-phosphate, G2P glycerate 2-phosphate, GAPDH glyceraldehyde-3-phosphate dehydrogenase, PGK phosphoglycerate kinase, PGM phosphoglycerate mutase, ENO enolase, MDH malate dehydrogenase, FUM fumarate hydratase, FRD fumarate reductase, SUC succinyl-CoA synthetase, MUT methylmalonyl-CoA mutase, MCEE methylmalonyl-CoA epimerase, PCC propionyl-CoA carboxylase, PCT propionate CoA transferase, PK pyruvate kinase, PFL pyruvate formate-lyase, KOR 2-oxoglutarate/2-oxoacid ferredoxin oxidoreductase, HADH 3-hydroxyacyl-CoA dehydrogenase, ECH enoyl-CoA hydratase, ENR enoyl-[acyl-carrier protein] reductase, PTB phosphate butyryltransferase, BUK butyrate kinase, BCoAT butyryl CoA: acetate CoA transferase. **c** Sequence alignment of identified acetylated PCK (MH0173_GL0113524, Kac peptide GFTAKacLAGTER) with known PCKs in PDB database. Taxonomic origin and starting amino acid position are indicated in the left side. The consensus sequence is colored in blue gradient according to the percentage identity. *A. succinogenes Actinobacillus succinogenes, E. coli Escherichia coli, T. thermophiles Thermus thermophiles, T. cruzi Trypanosoma cruzi*. **d** GTP-dependent and ATP-dependent PCKs share the same catalytic structural elements. The structure of the catalytic pocket of the GTP-dependant rat PCK (colored in gray, PBD 3DT4) is superposed with *A. succiniciproducens* ATP-dependant PCK (colored in Cyan, PBD 1YTM). Three catalytic elements: R loop, P loop, and Ω-loop are highlighted with light blue, light red, and light yellow, respectively, in rat PCK, and with bright blue, bright red, and bright yellow, respectively, in *A. succiniciproducens* PCK. The oxalate and ATP are indicated as sticks and colored by atom type. The Mg and Mn metals are indicated as green spheres. **e** Interaction among K384, E389, R60, and oxalate in *A. succiniciproducens* PCK. Protein structure was generated with PyMOL (https://pymol.org/).

carbohydrate metabolism was the most widely acetylated metabolic pathway in microbiome, in particular pyruvate metabolism (49 KOs), amino sugar and nucleotide sugar metabolism (46 KOs), glycolysis/gluconeogenesis (41 KOs), fructose and mannose metabolism (39 KOs), butanoate metabolism (36 KOs), propanoate metabolism (35 KOs), and starch and sucrose metabolism (35 KOs) (Supplementary Data 5). In addition, we established complete anaerobic fermentation pathways that produce SCFAs using the identified Kac enzymes in this study (Fig. 3b), indicating a widespread protein acetylation of the enzymes involved in these important gut microbial metabolic pathways.

Examining the top ten most abundant Kac peptides identified in this study, we found that nine were from bacteria and one from human chymotrypsin-like elastase family member 3A (Supplementary Table 4). Moreover, all the nine most abundant microbial Kac peptides were from enzymes involved in the SCFA production, including glyceraldehyde-3-phosphate dehydrogenase

(GAPDH, three peptides), 3-phosphoglycerate kinase (PGK, two peptides), pyruvate: ferredoxin oxidoreductase (PFOR, two peptides), phosphoenolpyruvate carboxykinase (ATP-dependent) (PCK, one peptide), and 3-hydroxyacyl-CoA dehydrogenase (HADH, one peptide) (Fig. 3b). Six of the nine microbial Kac peptides had the lowest-common ancestor (LCA) of bacteria or root (shared by all organisms), while the other three were unique to species belonging to *Clostridiales* (Supplementary Table 4), the major SCFA producers in microbiota.

We then took PCK as an example to examine whether the identified Kac site is important for regulating enzymatic activity. PCK catalyzes the reversible conversion of phosphoenolpyruvate (PEP) into oxaloacetate (Fig. 3b). A blastp search against NCBI-nr database revealed high sequence similarity of PCKs across different bacterial species (>89% identity and >98% coverage for top 100 hits; >99% hits were from Firmicutes). Alignment of the identified PCK protein sequence in this study to known PCK proteins in Protein Data Bank (PDB) identified *Anaerobiospirillum*

*succiniciproducens* PCK as the most similar one (full-length protein sequence identity of 82% and *E*-value of 1E-81) (Fig. 3c). There are three essential catalytic structural elements in PCK, namely P loop, R loop, and Ω loop (Fig. 3d). P loop and the R loop are directly involved in catalysis and substrate binding, while Ω-loop act as a lid gate by switching from a closed-active conformation to an open-inactive conformation[35,36]. *E. coli* PCK with the truncated or shortened Ω-loop has been reported to loss enzyme activity[37]. In the crystal structure of *A. succiniciproducens* PCK with a closed-active conformation, the mapped site K384 at Ω-loop forms a salt bridge with glutamine 389 (E389) which interacts with key catalytic residue arginine 60 (R60) located at R loop (Fig. 3e). Therefore, the acetylation of K384 could interrupt these interactions and de-stabilize the active conformation of PCK. Further biochemical validation is still required, however, this finding suggests that the exploration of the current data set helps the study of how protein acetylation may regulate gut microbial activity.

**Alterations of gut microbial lysine acetylome in CD patients.** To demonstrate the applicability of the integrated lysine acetylomic/metaproteomic approach, we analyzed intestinal MLI aspirate samples collected from pediatric CD patients. We first analyzed the lysine acetylomes of time-series MLI aspirate samples collected from three pediatric CD patients who were undergoing disease alleviation following treatment (Supplementary Table 5). In total, 10,085 protein groups and 9225 Kac sites

were identified for metaproteomic and lysine acetylomic aliquots, respectively. PCA analysis showed that samples collected from the same patients clustered together for both metaproteome and lysine acetylome, as well as their ratios, albeit collected up to 46 months apart, and with disease alleviation (Supplementary Fig. 5). This result suggests that both the metaproteome and lysine acetylome of an individual's microbiome are relatively stable over time. This is in agreement with previous metagenomic studies showing long-term stability of both functional- and strain-level compositions of gut microbiota[38,39]. We then analyzed 18 intestinal aspirate samples collected from treatment-naive patients, including 10 pediatric CD patients (male/female, 6/4; age, 13.1 ± 2.1) and 8 control subjects (male/female, 2/6; age, 13.0 ± 4.5) (Supplementary Table 6). In total, we accurately quantified 4623 Kac sites in lysine acetylomic aliquot and 17,684 protein groups in metaproteomic aliquot. Principal component analysis (PCA) showed a trend to separate both the metaproteome and the lysine acetylome for CD versus control (Fig. 4a, b). Partial least square discriminant analysis (PLS-DA) was then used for supervised group classification and identifying key variables in the metaproteome ($Q^2 = 0.59$), which identified 208 increased and 284 decreased protein groups in CD compared with control subjects (Supplementary Data 6). PLS-DA of the lysine acetylome ($Q^2 = 0.33$) identified 51 increased and 29 decreased Kac sites in CD compared with control subjects (Supplementary Data 6). In addition to those PLS-DA-identified differentially abundant proteins or Kac sites, the proteins and Kac sites that were detected

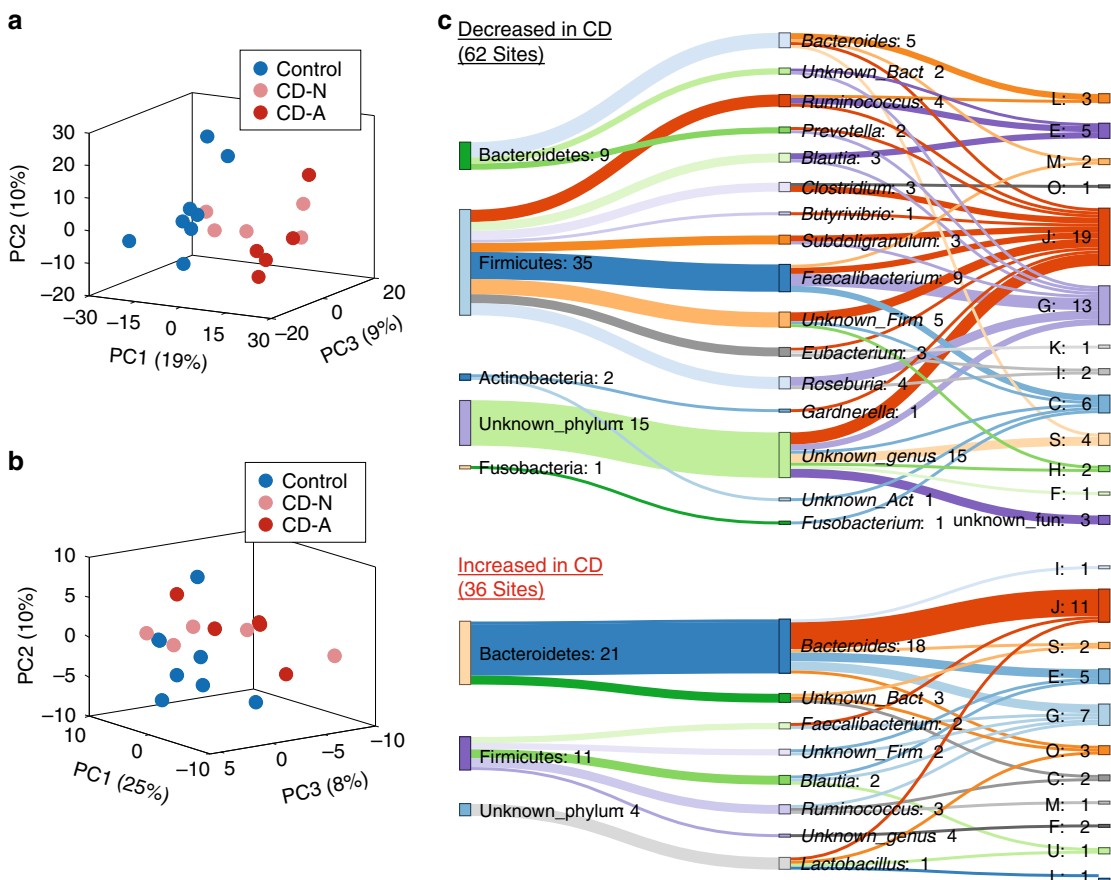

**Fig. 4 Lysine acetylome alterations of the intestinal microbiome in pediatric CD. a** PCA score plot of the metaproteome of the intestinal microbiome. **b** PCA score plot of the lysine acetylome of the intestinal microbiome. **c** Differentially abundant microbial Kac sites. The COG category and taxonomy (phylum and genus) for the differentially abundant Kac sites are shown in the Sankey plot. The numbers after the colons indicate the numbers of differentially abundant Kac sites. The phylum-genus links and genus-function (COG category) links are shown. Each letter corresponds to a COG category as shown in Fig. 3. The Sankey plot was generated using SankeyMATIC (http://sankeymatic.com/). Source data are provided as a Source Data file.

in ≥75% of samples in the one group and ≤25% of samples in the other group were also considered as differentially abundant. Altogether, the current study identified 82 Kac sites that were increased, and 68 Kac sites that were decreased, in CD compared with control subjects (Supplementary Data 6).

Among the 82 Kac sites that were increased in CD compared with control subjects, 46 were from human proteins and 36 were from microbiome proteins. However, only six Kac sites that were decreased in CD were from human proteins, and the remaining 62 downregulated Kac sites were from microbiome proteins. Interestingly, 11 and 21 upregulated microbial Kac sites, while 35 and 9 downregulated microbial Kac sites were derived from Firmicutes and Bacteroidetes, respectively (Fig. 4c). The majority of the 35 downregulated Firmicutes-derived Kac sites were from known SCFA-producing bacteria, including *Faecalibacterium* (nine sites), *Ruminococcus* (four sites), *Roseburia* (four sites),

*Eubacterium* (three sites), *Subdoligranulum* (three sites), *Clostridium* (three sites), and *Blautia* (three sites). Taxon-specific functional analysis showed that the microbial Kac sites that showed increased abundances in CD were mainly from translation-related proteins of *Bacteroides*; and the downregulated microbial Kac sites in CD were mainly from proteins that are involved in translation and carbohydrate metabolism of known SCFA producers as mentioned above (Fig. 4c).

We also performed comparative taxonomic analysis using the quantified Kac microbial peptides as well as non-Kac peptides in metaproteomic aliquots with linear discriminant analysis effect size (LEfSe) analysis. The results showed that the acetylome-based abundances of species *Roseburia inulinivorans*, *Eubacterium eligens*, and *Megamonas funiformis* were significantly decreased in CD compared with that of controls, and the abundance of *Bacilli* was significantly increased (Fig. 5a). Metaproteome-based

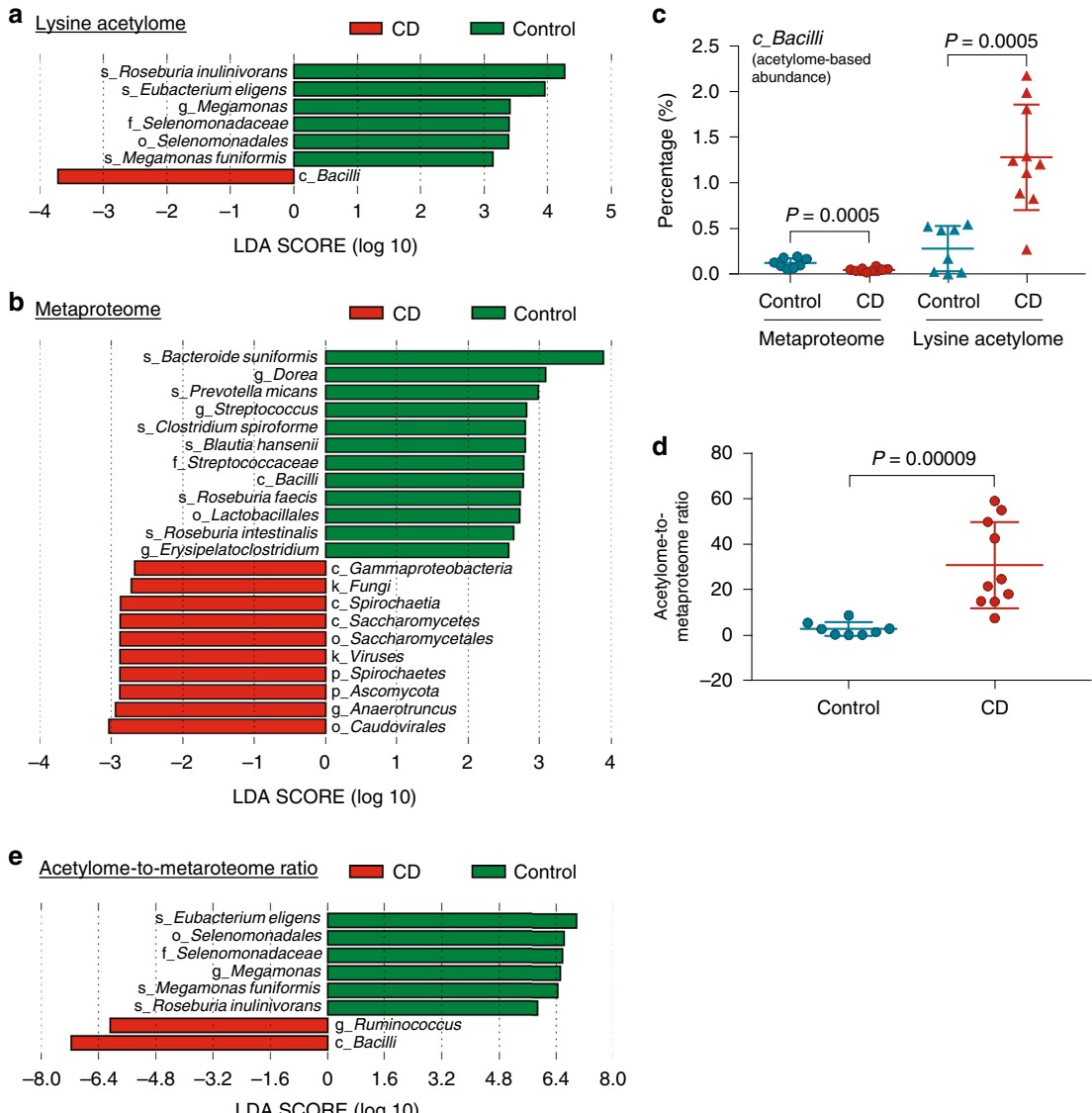

**Fig. 5 Taxonomic alterations of protein acetylation in the pediatric CD microbiome. a** LEfSe analysis of lysine acetylome-based taxonomic compositions. **b** LEfSe analysis of metaproteome-based taxonomic compositions. **c** Percentage of *Bacilli* in metaproteome and lysine acetylome data sets. Control, *n* = 8 biologically independent samples; CD, *n* = 10 biologically independent samples. Statistical significance of the difference between groups was evaluated using two-sided Mann–Whitney *U* test. **d** Acetylome-to-metaproteome ratios of *Bacilli* in pediatric CD and control subjects. Control, *n* = 8 biologically independent samples; CD, *n* = 10 biologically independent samples. Statistical significance of the difference between groups was evaluated using two-sided Mann–Whitney *U* test. **e** LEfSe analysis of the acetylome-to-metaproteome ratios of all quantified taxa in the lysine acetylome data set. For scatter dot plot, mean (long line) and standard deviation (short line) are indicated. Source data are provided as a Source Data file.

taxonomic analysis identified 12 taxa that were decreased and 10 taxa that were increased in CD compared with control subjects (Fig. 5b). Interestingly, the metaproteome-based abundance of *Bacilli* was decreased in CD, which is opposite to the observations in lysine acetylome (Fig. 5c). Accordingly, the ratios of acetylome-based abundance to metaproteome-based abundance of *Bacilli* were significantly increased in CD compared with controls (*P* < 0.0001, Fig. 5d), highlighting the additional information provided by lysine acetylome in this study. LEfSe analysis using the acetylome-to-metaproteome ratios of all 103 quantified taxa identified six taxa that exhibited significantly decreased ratios in CD compared to control, and two taxa (*Bacilli* and *Ruminococcus*) that exhibited increased ratios (Fig. 5e). Evaluation of the six abundant bacterial phyla revealed that, compared to control subjects, CD patients displayed higher acetylome-to-metaproteome ratios for Actinobacteria, Bacteroidetes and Proteobacteria, while lower ratios for Firmicutes (Supplementary Fig. 6). Interestingly, we found that the acetylome-to-metaproteome ratios of these abundant phyla were all partially reverted during disease alleviation, in particular for the first post-treatment time point when all three patients were in remission

(Supplementary Fig. 7). *Bacilli* and *Ruminococcus* also showed a trend of decreased acetylome-to-metaproteome ratios when the patients were in remission in the first post-treatment time point (Supplementary Fig. 7).

**Altered microbiome-associated human protein Kac levels in CD.** As mentioned above, we identified 46 human protein Kac sites that showed increased abundances and six human protein Kac sites that showed decreased abundances in CD compared to control (Fig. 6). The increased human protein Kac sites were mainly from the proteins lactotransferrin (LTF; ten sites) and calprotectin (protein S100A8 and S100A9; seven sites). In addition, there were 11 elevated protein Kac sites from immunoglobulins (Ig), including Ig heavy-constant alpha (IGHA1 and IGHA2), Ig heavy-constant mu (IGHM), Ig lambda constant 3 (IGLC3), Ig kappa constant (IGKC), and Ig lambda-like polypeptide 5 (IGLL5). Calculating the ratios between each of the 52 differentially abundant Kac sites and their corresponding protein abundances in the metaproteomic aliquot revealed that 27 out of the 52 Kac sites exhibited significantly different site-to-protein

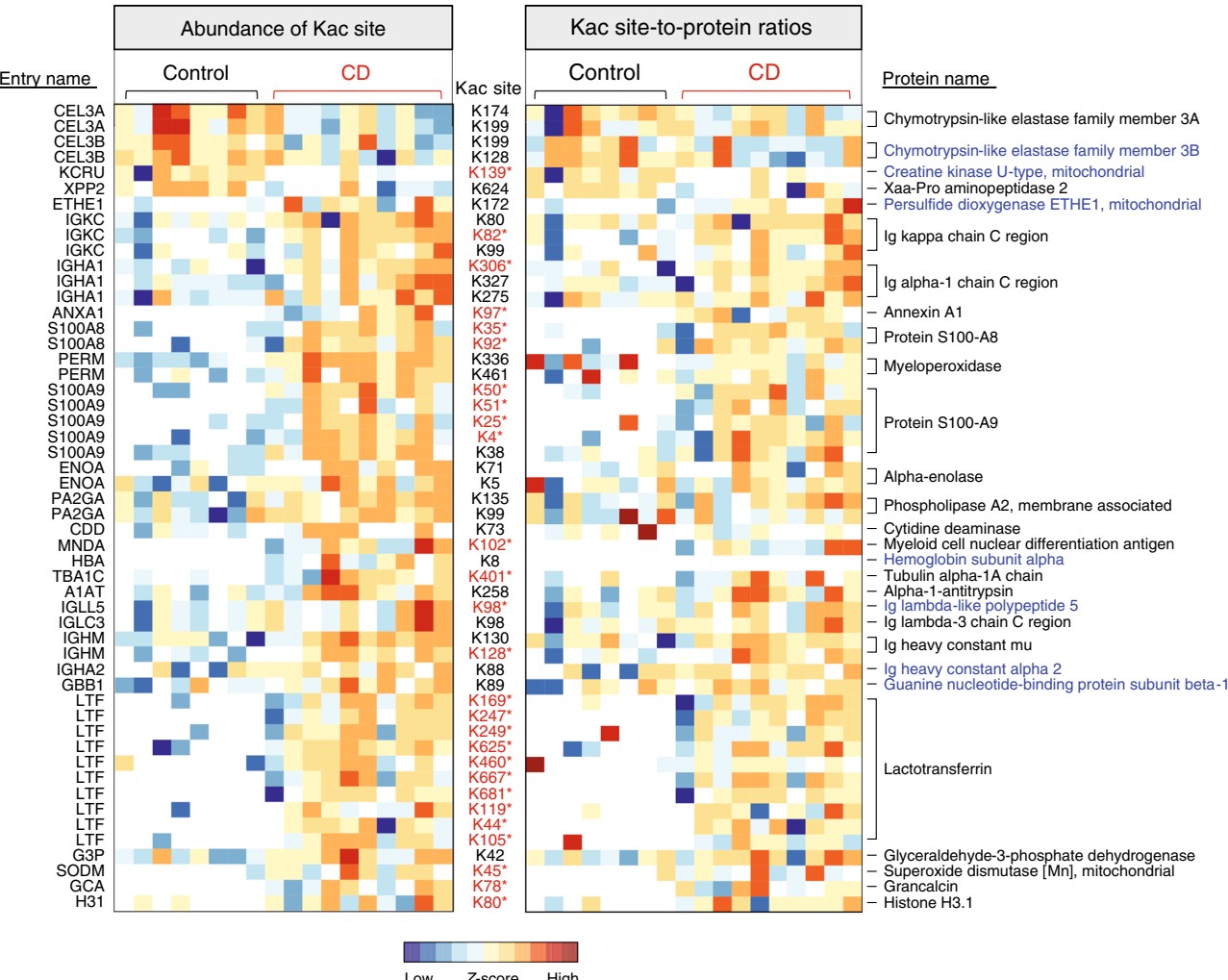

**Fig. 6 Abundance alterations of human protein Kac sites in CD microbiome samples.** A heatmap of differentially abundant human protein Kac sites is shown on the left, and a Kac site-to-protein ratio heatmap is shown on the right. Each row of the heatmap is a protein Kac site (indicated in between the two panels). The UniProt protein entry name and protein name for each Kac site are indicated on the left side and right side, respectively. The Kac sites highlighted in red stars retained the differences in their site-to-protein ratios. Protein names highlighted in blue indicate the proteins with no significant difference between CD and control in unenriched samples. The heatmap was generated using iMetaLab (http://imetalab.ca/). Source data are provided as a Source Data file.

ratios between control and CD patients (Mann–Whitney test, $P < 0.05$) or frequencies of detection.

In this study, we found that the total protein levels of both S100A8 and S100A9, two monomers of a known CD biomarker calprotectin[40,41], were significantly increased in CD compared with controls (Supplementary Fig. 8a). In addition, we identified five Kac sites for protein S100A8 and seven Kac sites for protein S100A9 in the lysine acetylome data set, among which two for S100A8 and five for S100A9 were detected in <20% of control samples and >80% of CD samples (Supplementary Fig. 8b, c). Among all the identified calprotectin Kac sites, only one site for S100A8 (K18) and one site for S100A9 (K38) were quantified in ≥3 control samples, and both showed significantly lower abundances in control than CD. Although there is no significant difference on the site-to-protein ratios of individual Kac site between CD and control samples, the ratios between the sum intensities of all Kac sites on S100A8, S100A9, and their corresponding proteins were significantly higher in CD compared with control samples (S100A8, $P = 0.02$; S100A9, $P = 0.02$) (Supplementary Fig. 8d). Evaluating the changes of Kac sites of S100A8 and S100A9 following disease alleviation in the treatment cohort, we found that the overall Kac levels of both S100A8 and S100A9 showed decreasing trend after treatment (Supplementary Fig. 9). Among the Kac sites identified in this therapeutic cohort, a general decreasing trend was observed, in particular for those that have been identified as being significantly increased in CD compared to control, such as S100A8 K18, K35, S100A9 K4, and K38 (Supplementary Fig. 9). These findings further validate the alterations of lysine acetylome identified when comparing CD with controls.

## Discussion

Lysine acetylation is an important PTM event regulating various biological processes and cellular functions in all kingdoms of organisms[7,8,42,43]. The global profiling of Kac has been performed in many organisms, including bacterial species such as *Escherichia coli*[13,16–18]. However, as mentioned above, the global characterization of Kac sites in the microbiome has not yet been studied. This is mainly due to the extremely high complexity of microbiome samples, which requires an enrichment approach with better coverage, and due to the bioinformatic challenges in efficiently identifying and quantifying the microbiome Kac peptides. In this study, we utilized the seven anti-Kac monoclonal antibody cocktail, which was developed by Svinkina et al.[27], for the enrichment of Kac peptides from tryptic digest of microbiome proteins. In addition, we developed an integrated metaproteomics/lysine acetylomics data-processing workflow based on our previously developed MetaPro-IQ approach[28] and MetaLab software tool[29]. Altogether, the experimental and bioinformatic workflow enabled a successful Kac peptide enrichment, identification, and quantification for microbiome samples. In total, over 35,000 Kac peptides were identified in enriched samples, which is far higher than that in unenriched samples (117 Kac peptides). It is worth noting that only 6 out of the 117 Kac peptides that were identified in unenriched samples overlapped with those identified in the enriched samples. Given that only 0.2% of the peptides identified in unenriched aliquot were Kac peptides (less than the FDR threshold of 1% for target-decoy database search) and their lower peptide-spectrum match (PSM) scores (Supplementary Fig. 10), the Kac peptides identified from the unenriched aliquot were potentially false identifications. This finding further suggests that an enrichment step during sample preparation is necessary to deeply and reliably identify protein lysine acetylation in the microbiome.

In bacteria, nonenzymatic acetylation by acetylphosphate has been considered as a major contributor for protein acetylation[13,44]. In enzymatic acetylation mechanism, a catalytic glutamate (E) residue in the enzyme is required to deprotonate the epsilon–amino group of the target lysine[12]. Similarly, an internal acidic amino acid, such as E or D, near the target lysine is required to deprotonate the epsilon–amino group in a nonenzymatic mechanism[12]. Accordingly, we found that the −1 position of microbiome Kac site was significantly enriched by E and D (top 1 and 2, respectively; Fig. 1f), suggesting that nonenzymatic acetylation mechanism is predominantly present in the gut microbiome. In addition, the relative abundance of enzymes for the generation of acetylphosphate from acetyl-CoA significantly correlated with the overall Kac levels in microbiome samples, while ACAT (converts acetyl-CoA to acetoacetyl-CoA for the production of butyrate) negatively correlated with the overall Kac levels. These findings provide evidence for a nonenzymatic protein acetylation mechanism in prokaryotes at the microbiome level.

Acetylphosphate is a key metabolic intermediate in acetate metabolism, abundantly present in SCFA producers in gut microbiota. Firmicutes is one of the most abundant bacterial phyla and major SCFA-producing bacteria, which plays important roles in human health at least in part through generating SCFAs and harvesting energy from indigestible dietary fibers[45,46]. Accordingly, nearly half of the identified Kac peptides in both adult stool and pediatric MLI aspirate samples were derived from Firmicutes. This is also in agreement with the observations that most acetylphosphate-generating enzymes identified in this study are derived from Firmicutes, and the latter had higher lysine acetylome-to-metaproteome ratios than other bacterial phyla. Interestingly, we found that Kac is a common PTM event for almost all the important enzymes in SCFA metabolism in gut microbiome, which may be due to non-enzymatically acetylation by the excessive acetylphosphate within the cellular compartment. Castano-Cerezo et al. previously reported that many proteins involved in acetate metabolism, including ACS which converts acetate to acetyl-CoA, are acetylated proteins and their activities are also regulated by lysine acetylation[18]. In *Salmonella*, Wang et al. demonstrated that enzymes in central metabolic pathways were extensively acetylated and protein acetylation regulated the direction of carbohydrate metabolic flux in response to environmental changes[8]. In this study, the structural analysis of PCK, one of the most abundant Kac enzymes, also suggested that acetylation might be involved in regulating the direction of SCFA metabolism. We found that the catalytically essential structure loop of PCK, which regulates enzyme conformation[35,36], was among the most abundantly acetylated proteins in the microbiome. The acetylation of K473 in rat PCK loop, which shares highly similar secondary structure to that of bacterial PCK (Fig. 3d), has been shown to significantly increase the efficiency of conversion from phosphoenolpyruvate (PEP) into oxaloacetate, while decrease the efficiency of gluconeogenic reaction (oxaloacetate to PEP)[47]. This suggests that the identified Kac site on gut microbial PCK might be involved in accelerating the metabolic flow of PEP to oxaloacetate and thereby succinate/propionate (Fig. 3b). Taken together, these findings suggest that gut microbial Kac might be an important mechanism regulating the SCFA metabolism and influences the complex host–microbiome interactions in diseases.

Intestinal dysbiosis, in particular the depletion of SCFA-producing bacteria, is commonly associated with the development of both adult and pediatric CD[25,48,49]. In this study, we demonstrated that the intestinal dysbiosis observed in pediatric CD patients includes alterations in the lysine acetylation of microbiome proteins, in particular the decreased Kac levels of proteins of butyrate/acetate-producers. As mentioned above, lysine acetylation may act as a regulating factor for SCFA production in gut microbiome. Although further studies are still needed to understand whether the decrease of protein Kac levels may contribute

to the depletion of SCFA producers or not in CD patients, the findings in this study indicate that lysine acetylation might be a potential target for manipulating the growth and functional activity of SCFA producers, and thereby for the treatment of diseases such as CD. Accordingly, previous studies have shown that lysine deacetylase (KDAC) inhibitors, such as butyrate, suberyolanilide hydroxamic acid (SAHA), valproic acid (VPA), and statin hydroxamate, effectively treat colitis in animal models[50–53]. In particular, Wei et al. reported that statin hydroxamate alleviated colitis and reduced the blood endotoxin (lipopolysaccharide) level[51], an indicator of dysbiosis of gut microbiota[54]. Although the mechanism has been attributed to their beneficial effects on the intestinal epithelium cells, our findings indicate that the KDAC inhibitors may in part directly interact with the microbiome for the alleviation of intestinal colitis. Further studies examining the microbiota composition and microbial lysine acetylome of animals or patients after KDAC inhibition treatment would be worthwhile.

To conclude, we demonstrated that a recently developed, commercially available anti-Kac monoclonal antibody cocktail can be used to successfully enrich Kac peptides from the human microbiome. Based on this, we established an experimental and bioinformatic data-analysis workflow for microbiome-wide characterization of protein lysine acetylation. We revealed widespread protein lysine acetylation in important metabolic pathways of human gut microbiota, in particular those for producing SCFAs in Firmicutes. Analyzing the microbiome samples collected from the intestinal mucosal surface of pediatric CD and control subjects revealed that the majority of downregulated Kac sites in CD patients belonged to the SCFA-producing bacteria in Firmicutes. In addition to the microbiome Kac sites, we also identified Kac sites on human proteins that were associated with the microbiomes and demonstrated the alterations of Kac levels on immune response proteins, such as immunoglobulins and calprotectin. This study was limited by the number of CD patients, however, the findings provide valuable information for designing further studies to understand the functionality of the microbiome in CD.

## Methods

**Subjects and sample collections**. Fresh fecal samples were collected from six healthy adult volunteers at the University of Ottawa with protocol (Protocol # 20160585-01H) approval by the Ottawa Health Science Network Research Ethics Board at the Ottawa Hospital. Informed consent was obtained from all adult subjects. Descending colon MLI aspirate samples were collected from pediatric patients that were undergoing initial diagnostic evaluation for possible CD and then subsequent colonoscopy required for their medical care. The study was approved by the Research Ethics Board of the Children's Hospital of Eastern Ontario (CHEO), Ottawa, ON, Canada. Written informed consent form was obtained from their parents. The study complies with all relevant ethical regulations for research with human participants.

At diagnosis, all participants (<18 years old) were treatment-naive. CD was diagnosed through clinical, endoscopic, histologic, and radiological evaluations according to standard criteria[55]. Disease severity was determined using the Pediatric Crohn's Disease Activity Index (PCDAI)[56]. The Simplified-Endoscopy Score-Crohn's disease (SES-CD) was used as segmental description of colonoscopic mucosal characteristics (i.e., presence and size of ulcers, extent of ulcerated surfaces, extent of affected surfaces, and presence and severity of luminal narrowing) with each characteristic being scored from 0 to 3[57]. Control patients had visually normal mucosa, histologically normal mucosal biopsies and normal imaging. The following exclusion criteria were implemented to further refine the cohort enrolled in this study: (1) presence of diabetes mellitus; (2) presence of infectious gastroenteritis within the past 2 months; (3) use of any antibiotics or probiotics within the past 4 weeks, or (4) irritable bowel syndrome. Patient clinical data were collected and managed using REDCap (Research Electronic Data Capture) hosted at the CHEO Research Institute. REDCap is a secure, web-based application designed to support data capture for research studies[58]. The MLI aspirate samples were collected, as previously described[3,25]. Briefly, any existing fluid and debris were first aspirated and discarded during colonoscopy. Sterile water was then flushed onto the mucosal surface to dislodge the mucus layer from the epithelial cells; the resulting fluid was then aspirated into a container. The latter was immediately put on ice and transferred to the laboratory for further processing.

**Sample processing, protein extraction, and tryptic digestion**. The fresh stool sample was immediately put on ice and subjected to differential centrifugation, and washed according to the procedures described previously[59]. The resulting microbial pellets were then subjected to protein extraction using lysis buffer containing 4% (w/v) SDS, 50 mM Tris-HCl (pH 8.0), and protease inhibitor (cOmplete™, mini protease inhibitor cocktail; Roche Diagnostics GmbH). Protein lysates were then precipitated by adding five times the volume of lysis buffer of ice-cold acidified acetone/ethanol buffer overnight at −20 °C. Precipitated proteins were then collected with centrifugation at 16,000 g for 25 min at 4 °C, and washed three times by ice-cold acetone before re-suspending in 6 M urea, 100 mM ammonium bicarbonate buffer[3]. Protein concentration was determined using a DC protein assay (Bio-Rad Laboratories, Inc) according to the manufacturer's instruction. Approximately 10–15 mg of proteins for each sample were then used for in-solution proteolytic digestion. Briefly, proteins were first reduced with 10 mM dithiothreitol (DTT) for 1 h and 20 mM iodoacetamide (IAA) for 40 min at room temperature; then the samples were diluted by tenfold with 100 mM ammonium bicarbonate buffer followed by digestion using lysyl endopeptidase (Lys-C; Wako Pure Chemical Corp., Osaka, Japan) for 4 h and trypsin (Worthington Biochemical Corp., Lakewood, NJ, USA) for overnight at room temperature. The resulting digests were then subjected to desalting using Waters Sep-Pak® Vac 3cc (200 mg) tC18 cartridges and eluted using 80% acetonitrile/0.1% formic acid. A small portion of the proteolytic peptides from each sample was used for metaproteomic analysis (directly load for MS analysis), and the remainder was used for Kac peptide enrichment.

For MLI aspirate samples, upon arriving at the laboratory, the samples were immediately mixed with protease inhibitor (cOmplete™, mini protease inhibitor cocktail; Roche Diagnostics GmbH). The aspirate samples were first centrifuged at 700 g for 5 min at 4 °C, and the supernatant collected for another centrifugation at 14,000 g for 20 min at 4 °C. The pellet fraction was harvested for protein extraction using lysis buffer consisting of 4% (w/v) SDS, 8 M urea, 50 mM Tris-HCl (pH 8.0), and cOmplete™ mini protease inhibitor cocktail. Protein lysates were then precipitated and washed using ice-cold acetone as described above. An equal amount (2.5 mg) of proteins for each sample was then used for in-solution trypsin digestion and desalting using Waters Sep-Pak® Vac 3cc (200 mg) tC18 cartridges as described above. A small portion of the tryptic peptides (equivalent to 40 µg proteins) from each sample was used for metaproteomic analysis (directly load for MS analysis), and the remainder was used for Kac peptide enrichment.

**Kac peptide enrichment**. Kac peptides were enriched using PTMScan® Motif antibody kits (Cell Signaling technology, Inc.) according to the manufacturer's instruction. Briefly, tryptic peptides were first re-suspended in PTMScan® IAP Buffer and centrifuged at 10,000 g for 5 min at 4 °C to remove any insoluble pellets. The supernatant was then added directly to the tube containing Kac motif antibody beads and mixed immediately by slowly pipetting up and down. The mixture was then incubated at 4 °C for 2 h on a rotator. After incubation, the beads were collected by centrifugation at 2000 g for 30 s. The beads were then washed twice with cold IAP buffer and three times with H₂O. The peptides were eluted by adding 55 µl of 0.15% (v/v) trifluoroacetic acid (TFA) to the beads and incubating for 10 min while mixing gently. The supernatant was collected through centrifuging at 2000 g for 30 s, and the remaining beads were mixed with another 50 µl of 0.15% (v/v) TFA for another round of elution. Both eluents were then combined for desalting using 10-µm C18 columns (5 mg per column). After two washes using 0.1% formic acid, peptides were eluted using 80% acetonitrile/0.1% formic acid and evaporated with a Speed-Vac concentrator for mass spectrometry analysis[59].

**Tandem mass spectrometry analysis**. The peptides generated from metaproteomic and lysine acetylomic aliquotes of fecal microbiome samples were analyzed using Q Exactive HF-X mass spectrometer (ThermoFisher Scientific Inc.). For unenriched samples, peptides equivalent to 250 ng proteins were loaded for MS analysis; for enriched samples, all peptides were re-suspended in 20 µl 0.1% (v/v) formic acid, and 4 µl was used for MS analysis. Peptides were separated on an analytical column (75 µm × 15 cm) packed with reverse-phase beads (1.9 µm; 120-Å pore size; Dr. Maisch GmbH, Ammerbuch, Germany) with 2 h gradient from 5 to 35% (v/v) acetonitrile at a flow rate of 300 nl/min. The instrument method consisted of one full MS scan from 350 to 1400 m/z followed by data-dependent MS/MS scan of the 16 most intense ions and a dynamic exclusion duration of 20 s. Mass spectrometry analysis of MLI aspirate samples, including both enriched and unenriched samples, was performed on a Q Exactive mass spectrometer (ThermoFisher Scientific Inc.). For unenriched samples, peptides equivalent to 1 µg proteins were loaded for MS analysis; for enriched samples, all peptides were re-suspended in 20 µl 0.1% (v/v) formic acid, and 4 µl was used for MS analysis. Briefly, peptides were separated on an analytical column (75 µm × 15 cm) packed with reverse-phase beads (1.9 µm; 120-Å pore size; Dr. Maisch GmbH, Ammerbuch, Germany) with 4 h gradient from 5 to 35% (v/v) acetonitrile at a flow rate of 300 nl/min. The instrument method consisted of one full MS scan from 300 to 1800 m/z followed by data-dependent MS/MS scan of the 12 most intense ions, a dynamic exclusion repeat count of 2, and repeat exclusion duration of 30 s. The MS data were recorded with the Thermo Xcalibur™ software (version 3.1) and exported in RAW format for further bioinformatic data processing.

**Identification of Kac and non-Kac peptides and proteins**. Protein identification and quantification for both metaproteomic and lysine acetylomic data sets were performed using MetaLab[29] with a modified MetaPro-IQ workflow[28] as detailed in Fig. 1b. In this study, we used the human fecal microbial Integrated Gene Catalog database (IGC; downloaded from China National GeneBank https://db.cngb.org/microbiome/genecatalog/genecatalog_human/) for adult stool samples and the protein database that was generated in our previous metaproteomic studies of pediatric MLI aspirate samples[3], which we termed the IGC + database in this study, for MLI aspirate samples. Briefly, the IGC + database consists of protein sequences from IGC database[30], NCBI viral proteins, predicted protein sequences from shotgun metagenomic sequencing of MLI aspirate samples, and representative fungal species (details in ref. [3]). The same database search parameters were used for both metaproteomic and lysine acetylomic data sets as follows: (1) up to four missed cleavages are allowed, (2) fixed modification includes cysteine carbamidomethylation, (3) potential modifications include methionine oxidation, lysine acetylation, and protein N-terminal acetylation, and (4) a parent ion tolerance of 10 ppm and a fragment ion tolerance of 20 ppm. The peptide and protein identification were performed with a false discovery rate (FDR) threshold of 0.01.

The identified peptides and protein groups in unenriched samples were obtained from the modificationSpecificPeptides.txt and proteinGroups.txt files, respectively. The identified Kac peptides and sites in enriched samples were obtained from the modificationSpecificPeptides.txt and Acetyl (K) Sites files, respectively. Only Kac sites with a localization probability of >0.75 were used for further analysis.

**Kac-motif analysis**. The sequences of amino acids surrounding Kac sites were analyzed and visualized using WebLogo (https://weblogo.berkeley.edu/)[60]. Sequence windows of 11 amino acids surrounding Kac site were created. pLogo (https://plogo.uconn.edu/)[33] was used to identify statistically overrepresented Kac motifs for the identified Kac sites. Briefly, sequence windows with six upstream and downstream amino acids surrounding the Kac site were extracted from the database and submitted to Motif-X for Kac-motif extraction. The total identified microbial proteins from one randomly selected unenriched sample were used as background for microbial protein Kac-motif extraction.

**Taxonomy and functional analysis**. Taxonomic annotation of both unmodified and Kac peptides was performed using Unipept 4.0 with the Equal I and L and Advanced misscleavage handling options allowed[34]. Gene ontology (GO) term and Enzyme Commission (EC) number annotations were directly exported from Unipept analysis (https://unipept.ugent.be/). For enriched samples, only Kac peptides or proteins with Kac peptides were used. KEGG annotation and metabolic module construction were performed using GhostKOLA (https://www.kegg.jp/ghostkoala/).

Linear discriminant analysis (LDA) effect size (LEfSe) analysis[61] was used to identify differentially abundant microbial taxa between control and CD at all taxonomic levels. Relative abundance of taxon was calculated at each taxonomic rank level, namely kingdom, phylum, class, order, family, genus, and species. For LEfSe analysis, all values were multiplied by 1,000,000 according to the user instructions and the taxa with a logarithmic LDA score >2.0 were considered to be significantly different between groups.

**Multivariate and statistical analysis**. PCA was used to demonstrate the inter-sample distance/clustering in a non-supervised manner. PLS-DA, a supervised multivariate statistical method, was used for modeling the group classification and identifying variables that drive such discriminations. PCA and PLS-DA were performed on quantified protein groups for unenriched samples and Kac sites for enriched samples. Briefly, the protein groups or Kac sites that were quantified in >50% of the samples were used and missing values were imputed using K-nearest neighbor (KNN) method ($K = 5$). PCA and KNN imputation were performed in MATLAB (The MathWorks Inc.). PLS-DA was performed using MetaboAnalyst 4.0[62], and the proteins or Kac sites that had a Variable Importance in Projection (VIP) value ≥1 and $P < 0.05$ (Mann–Whitney $U$ test) were considered to be significantly different between control and CD groups.

Statistical significance of the difference between groups was evaluated using Mann–Whitney $U$ test (nonparametric test of the null hypothesis which is suitable for data that does not pass normality test), unless otherwise indicated. Functional enrichment analysis of identified microbial Kac proteins was performed with hypergeometric probability analysis using the microbial proteins identified in unenriched samples as background. Briefly, each of the proteins identified in unenriched or enriched samples was annotated with a COG database (ftp://ftp.ncbi.nih.gov/pub/COG/COG2014/data) using DIAMOND[63] (default parameters, E-value = 0.001) and the best hit for each query was selected for COG category assignment. The numbers of proteins assigned to each COG category from Kac proteins and background were used for calculating the significance $P$ values of enrichment using *hygecdf* function and Benjamini–Hochberg-adjusted FDR values using *mafdr* function in MATLAB (The MathWorks Inc.).

**Reporting summary**. Further information on research design is available in the Nature Research Reporting Summary linked to this article.

## Data availability

All MS proteomics data that support the findings of this study have been deposited to the ProteomeXchange Consortium (http://www.proteomexchange.org) with the data set identifier PXD015482 and PXD013427. Source data are provided with this paper.

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

## Acknowledgements

This work was supported by funding from the Natural Sciences and Engineering Research Council of Canada (NSERC), the Government of Canada through Genome Canada and the Ontario Genomics Institute (OGI-114 & OGI-149), CIHR grant number GPH-129340 and MOP-114872. D.F. acknowledges a Distinguished Research Chair from the University of Ottawa. We acknowledge Ruth Singleton and Christine Figeys at the CHEO, Ottawa ON, for their help in collecting intestinal aspirate samples. We also thank Dr. Kendra Hodgkinson at the University of Ottawa for her help in editing the paper.

## Author contributions

D.F., A.S., D.M., and X.Z. designed the study. D.M. collected patient samples and clinical data. S.A.D. and K.W. pre-processed the samples. X.Z. performed the experiments. X.Z., Z.N., Y.Y., and J.C. performed data analysis. M.S. and C.F. provided materials and involved in discussion of the study design. X.Z., D.F., A.S., D.M., and J.M. wrote the paper. All authors participated in the data interpretation, discussion and edits of the paper.

## Competing interests

D.F., A.S., and D.M. have co-founded MedBiome, a clinical microbiomics company. C.F. and M.S. are employees of Cell Signaling Technology. The remaining authors declare no competing interests.
