## [Peer Review File · Nature Communications]

Reviewers' Comments:

Reviewer #1:

Remarks to the Author:

This paper is novel for presenting a large-scale look at human intestinal microbiome (in situ colonic mucosal wash) protein modification (K-Ac), which is an important frontier issue in microbiome ecology. The data is analyzed to describe microbial composition and taxon-associated gene state in this microbiome compartment for a set of control adults, and in a comparison of pediatric endoscopic patients nominally Crohn's disease or control. The technical pipeline is state of art, and overall it provides a number of interesting observations.

There are a few limitations of this study.

Major

1. The paper implies that Kac-modification is highlighting representation of organisms and their functions that cannot be observed by straight metaproteomics. However, it is possible that these features are concordant with whole metaproteomics, with just additional incremental change when the Kac subset is assessed. That is, the Kac changes are mainly reflecting simply the underlying shift in the microbiome composition and function, rather than something special about the Kac subset. This possibility is supported by the observation that many of the same differential organisms and genes reported in the literature by metagenomics and metatranscriptomics in the normal microbiome, or in healthy vs. IBD, are also reported here using the Kac subset. This really should be addressed and clarified, to provide context on what is being gained by Kac analysis. (a) A way to analytically do this is to provide description and comparisons first using the all-peptide metaproteomics data for taxon and gene representation. This analysis would provide context for the present dataset. (b) And/or, in the discussion the authors could systemically tabulate the Kac subset findings compare for how these compare to the metaproteomics (or metagenomics and metatranscriptomics)literature for controls and disease/healthy comparisons. As presented, it is possible that the differential Kac features may simply can't tell whether Kac fraction is telling us

Technical

2. The statistics are can be better presented. The paper presents many types of data processing and statistical methods, but these aren't justified (why the particular method is most suitable or even appropriate?): paired T-test (line 184 for F/B ratio- is this normally distributed data?), hypergeometric test (line 191, for COG distribution among Kac vs. unmodified data); PLS-DA (line 227, for patient/control comparisons. And in some cases, the particular analytic question and how it was analyzed was unclear (e.g., Mann-Whitney in line 292, referring to a list of features and hence uncertain comparisons of microbial taxa, gene sets, and subject subsets). A section in methods or supplemental information can be used to provide rationale/justification for the analytics, and more clarity where needed in the results-findings regarding statistics used.

3. Down-regulated genes in CD (p. 10, para. starting with 238). It refers to "upregulated" genes, but it seems that these should be termed "down-regulated"

Reviewer #2:

Remarks to the Author:

This manuscript outlines a detailed study of the level and nature of protein lysine acetylation in the human microbiome. The authors utilize an anti-acetyl-lysine (Kac) antibody enrichment strategy and tandem mass spectrometry approach to identify 35,200 Kac peptides corresponding to 31,821 Kac sites from the microbial or host 27 proteins in human gut microbiome samples. They conduct a detailed functional analysis on the protein types, and then apply this approach to the analysis of

pediatric Crohn's 31 disease (CD) patient microbiome, where they identified 52 host and 136 microbial protein Kac sites that 32 were differentially abundant in CD versus controls.

In general, this manuscript is well-written and logically designed. The technical approach is sound and systematic. However, there are some fairly serious issues that reduce enthusiasm, as listed below:

1. The title strongly suggests that this manuscript is primarily a "methods paper," as there is really no description of why acetylation could be important in disease? While this is okay, the authors might consider revising to better define a "science message" if one is available.
2. Abstract (line 25) and Introduction - of all the possible PTMs, why focus on acetylation? The rationale is very weak and vague, which diminishes the overall impact of the study. Are there expectations that more or less acetylation is associated with disease somehow?
3. Line 98 – the authors should discuss the rationale for using the integrated gut microbial gene catalog vs. a matched assembled metagenome of the exact samples should be discussed and defended. How do the authors know that the gene catalog adequately reflects the exact nature and identify of the microbiota in their patients, especially the diseased cohort?
4. Line 114 – the observation of only 6 Kac peptides common in the enrichment compared to the 117 observed in the metaproteome is confusing and concerning. This issue is revisited on line 363, where the authors suggest that Kac peptides in the unenriched samples are likely false identifications. This is confusing and raises serious concerns with respect to false positive control in the other samples, and whether many of those might be false as well?
5. Line 117 – it seems a bit surprising that 80% of all protein groups have acetylation – was this expected? In light of the point above, this raises issue about false positives here again?
6. Line 156 – Do the authors suspect that the highest number of acetylation sites simply reflect the fact that Firmicutes and Bacteroidetes are simply the most abundant? In that case, acetylation might not be related to microbiome condition.
7. Line 216 – is there a basis to expect protein acetylation to be important in CD patients? Why this particular focus group? What will the "differences" mean? Line 383 provides some speculation, but even this linkage is quite vague and suspect.
8. The general outcome of this work is a broad and descriptive summary of protein acetylation, but delivers minimal biological value about what these differences mean?
9. Line 417 – One of the most concerning and unsettling aspects of this study is the sample cohort. The number of patients are low but not unacceptable for such a study. The most serious concern is that protein acetylation is known to be dynamic and thus temporal aspects are critical here. The use of only one sample per patient makes this study without context – how do the authors know that collection of samples from the same patients at another time-point wouldn't produce dramatically different results. This could be especially true in the diseased patients. This study would have been more meaningful if the authors had conducted multiple time-point measurements on a more limited number of the samples.

Reviewer #3:

Remarks to the Author:

In this manuscript, Zhang and colleagues have performed a characterization of the acetylation sites present in metaproteomic samples, including changes occurring in IBD patients. They have

identified on the order of 30,000 acetylation sites in bacterial proteins, which they have mapped to bacterial taxonomic diversity and protein functional annotations. As expected, most acetylation maps to proteins that relate to translation and metabolism. Interestingly, by performing also a separate proteomic analysis the authors could determinate the ratio of acetylated vs. non-modified peptides for different taxonomic groups. Finally, they analysed a small number of IBD patients and controls and performed a proof-of-principle analysis of how acetylation changes in the microbial samples of these patients. I think the number of samples is too low but serves as a useful test case application. The large number of acetylation sites identified in the study serves as potential exciting resource to study protein regulation by acetylation. However, the authors don't really make an effort of exploring this resource. The taxonomic differences in ratio of proteome acetylation is an interesting aspect that is mentioned mostly in discussion but it seems to be the most interesting result of the manuscript and could be better analysed. As it currently stands, the manuscript is mostly a quick description of acetylation patterns and does not go very far in novel discoveries.

Major concerns

The authors have identified what it is likely to be the largest collection of microbial acetylation sites to date. However, very little is done with this resource beyond a brief description of the taxonomic diversity and biological processes connected with these peptides. Understanding how most of the acetylation regulates protein activity is an important open question and this resource could be a fantastic opportunity to study this problem. The authors should provide some examples of how this could be achieved to motivate others to explore this great resource they have generated. For acetylated sites that match proteins that exist across many bacterial genomes the authors could align the protein sequences and map the acetylation sites to the alignment in order to study regions that are often acetylated across a large number of proteomes/species. For a number of these, there will be structural models that can serve as homology templates to study the structural impact of mapped acetylation sites. Since many of these will be in metabolic enzymes there should be many enzymes that are present in many different species and a number of structural models available to represent these enzymes. For example, the authors could ask if acetylation sites are often near catalytic residues for example.

In my opinion, the most interesting aspect of the results presented were the potential differences in ratio of acetylated vs non-modified peptides along the taxonomic diversity studied. The authors speculated that this differences in ratios could be due to metabolic differences with some species being known producers of acetate. Acetate can be converted to acetyl-phosphate in some species which can in turn directly chemically acetylate proteins (Weinert et al. Mol Cell 2013). The authors could try to further analyse this association using isolate genomes or very complete metagenomic assembled genomes. Can the authors find a relationship between the enzymes encoded in the genomes of some of the taxonomic groups and their ratios of acetylated vs non-modified peptides ?

Minor

The authors need to make available in supplementary material the list of identified peptides with MS relevant associated statistics in a way that can facilitate their further analysis.

The motif analysis of acetylated peptides could be relegated to supplementary information or shown briefly as an additional panel in Figure 1.

The IBD results serve as an interesting proof-of-principle but the sample sizes are not large enough to identify variations that may still be meaningful. This should be addressed in the discussion. Figure 6 could as well be moved to supplementary and significant differences indicated in Figure 5.

Reviewers' comments:

Reviewer #1 (Remarks to the Author):

This paper is novel for presenting a large-scale look at human intestinal microbiome (in situ colonic mucosal wash) protein modification (K-Ac), which is an important frontier issue in microbiome ecology. The data is analyzed to describe microbial composition and taxon-associated gene state in this microbiome compartment for a set of control adults, and in a comparison of pediatric endoscopic patients nominally Crohn's disease or control. The technical pipeline is state of art, and overall it provides a number of interesting observations.

There are a few limitations of this study.

Major

1. The paper implies that Kac-modification is highlighting representation of organisms and their functions that cannot be observed by straight metaproteomics. However, it is possible that these features are concordant with whole metaproteomics, with just additional incremental change when the Kac subset is assessed. That is, the Kac changes are mainly reflecting simply the underlying shift in the microbiome composition and function, rather than something special about the Kac subset. This possibility is supported by the observation that many of the same differential organisms and genes reported in the literature by metagenomics and metatranscriptomics in the normal microbiome, or in healthy vs. IBD, are also reported here using the Kac subset. This really should be addressed and clarified, to provide context on what is being gained by Kac analysis. (a) A way to analytically do this is to provide description and comparisons first using the all-peptide metaproteomics data for taxon and gene representation. This analysis would provide context for the present dataset. (b) And/or, in the discussion the authors could systemically tabulate the Kac subset findings compare for how these compare to the metaproteomics (or metagenomics and metatranscriptomics) literature for controls and disease/healthy comparisons. As presented, it is possible that the differential Kac features may simply can't tell whether Kac fraction is telling us

Reply:

We thank the reviewer for this comment. To elaborate on the added value of Kac data, we have added comparisons between metaproteome- and lysine acetylome-based differentially abundant taxa in CD versus controls in the revised manuscript (Pages 9-10, Lines 282-288). We have also added a new figure in the main manuscript (Fig. 5). These comparisons were discussed in the supplementary section of our original submission, but have now been moved to the main manuscript.

In addition, we also calculated ratios between the taxonomic abundances based on lysine acetylome and metaproteome for each quantified taxon, which is a proxy of relative lysine acetylation level. A comparison of this lysine acetylome-to-metaproteome ratios between CD and control subjects has also been added to the revised the manuscript in Page 10 Lines 282-294, as follows:

“We also performed comparative taxonomic analysis using the quantified Kac microbial peptides **as well as non-Kac peptides in metaproteomic aliquots with** linear discriminant analysis effect size (LEfSe) analysis. The results showed that the acetylome-based abundances of species *Roseburia inulinivorans*, *Eubacterium eligens* and *Megamonas funiformis* were significantly decreased in CD compared to that of controls, and the abundance of *Bacilli* was significantly increased (Fig. 5a). **Metaproteome-based taxonomic analysis identified 12 taxa that were decreased and 10 taxa that were increased in CD compared to control subjects (Fig. 5b). Interestingly, the metaproteome-based abundance of *Bacilli* was decreased in CD, which is opposite to the observations in lysine acetylome (Fig. 5c). Accordingly, the ratios of acetylome-based abundance to metaproteome-based abundance of *Bacilli* were significantly increased in CD compared to controls ($P < 0.0001$, Fig. 5d), highlighting the additional information provided by lysine acetylome in this study. LEfSe analysis using the acetylome-to-metaproteome ratios of all 103 quantified taxa identified six taxa that exhibited significantly decreased ratios in CD compared to control, and two taxa (*Bacilli* and *Ruminococcus*) that exhibited increased ratios (Fig. 5e). Evaluation of the six abundant bacterial phyla revealed that, compared to control subjects, CD patients displayed higher acetylome-to-metaproteome ratios for Actinobacteria, Bacteroidetes and Proteobacteria, while lower ratios for Firmicutes (Supplementary Fig. 5).”**

Fig. 5 Taxonomic alterations of protein acetylation in the pediatric CD microbiome. (a) LefSe analysis of lysine acetylome-based taxonomic compositions; (b) LefSe analysis of metaproteome-based taxonomic compositions; (c) Percentage of *Bacilli* in metaproteome and lysine acetylome data sets; (d) Acetylome-to-metaproteome ratios of *Bacilli* in pediatric CD and control subjects; (e) LefSe analysis of the acetylome-to-metaproteome ratios of all quantified taxa in the lysine acetylome data set.

Supplementary Fig. 5 Lysine acetylome-to-metaproteome ratios of gut microbial phyla in CD and control patients. The bottom and top of the boxes are the first and third percentile, respectively. The middle line represents the median (50th percentile). Whiskers are drawn from the ends of the interquartile range (IQR) to the furthest observations within 1.5 times the IQR range. Outliers >1.5 times the IQR are indicated with black dots.

Technical

2. The statistics are can be better presented. The paper presents many types of data processing and statistical methods, but these aren't justified (why the particular method is most suitable or even appropriate?): paired T-test (line 184 for F/B ratio- is this normally distributed data?), hypergeometric test (line 191, for COG distribution among Kac vs. unmodified data); PLS-DA (line 227, for patient/control comparisons. And in some cases, the particular analytic question and how it was analyzed was unclear (e.g., Mann-Whitney in line 292, referring to a list of features and hence uncertain comparisons of microbial taxa, gene sets, and subject subsets).

A section in methods or supplemental information can be used to provide rationale/justification for the analytics, and more clarity where needed in the results-findings regarding statistics used.

Reply:

We regret our lack of clarity for these important technical aspects. According to the reviewer's suggestion, we have expanded the "*Multivariate and statistical analysis*" in Methods section in Page 19 Lines 585-604, as follows:

"PCA was used to demonstrate the inter-sample distance/clustering in a non-supervised manner. PLS-DA, a supervised multivariate statistical method, was used for modeling the group classification and identifying variables that drive such discriminations. ... PLS-

DA was performed using MetaboAnalyst 4.0⁶² and the proteins or Kac sites that had a Variable Importance in Projection (VIP) value ≥ 1 and $P < 0.05$ (Mann-Whitney U test) were considered to be significantly different between control and CD groups.

Statistical significance of the difference between groups was evaluated using Mann-Whitney U test (nonparametric test of the null hypothesis which is suitable for data that doesn't pass normality test), unless otherwise indicated. Functional enrichment analysis of identified microbial Kac proteins was performed with hypergeometric probability analysis using the microbial proteins identified in unenriched samples as background. Briefly, each of the proteins identified in unenriched or enriched samples was annotated with a COG category as described previously²⁸. The numbers of proteins assigned to each COG category from Kac proteins and background were used for calculating the significance P values of enrichment using *hygecdf* function and Benjamini-Hochberg adjusted FDR values using *mafdr* function in MATLAB (The MathWorks Inc.).”

We have also added more explanations for the purpose/rationale of statistics in the Results section for PLS-DA (Page 9 Lines 254-255) and Mann–Whitney U test (Page 11 Lines 319-322). We thank the reviewer for reminding that normality of F/B ratio data distribution should be tested prior to statistics. In the revised manuscript, this result has been replaced with more comprehensive taxonomic analysis (please see our replies above to the major comment).

3. Down-regulated genes in CD (p. 10, para. starting with 238). It refers to "upregulated" genes, but it seems that these should be termed "down-regulated"

Reply:

We thank the reviewer for pointing this mistake out. This has been corrected in the revised manuscript (Page 9 Line 266).

Reviewer #2 (Remarks to the Author):

This manuscript outlines a detailed study of the level and nature of protein lysine acetylation in the human microbiome. The authors utilize an anti-acetyl-lysine (Kac) antibody enrichment strategy and tandem mass spectrometry approach to identify 35,200 Kac peptides corresponding to 31,821 Kac sites from the microbial or host 27 proteins in human gut microbiome samples. They conduct a detailed functional analysis on the protein types, and then apply this approach to the analysis of pediatric Crohn's disease (CD) patient microbiome, where they identified 52 host and 136 microbial protein Kac sites that 32 were differentially abundant in CD versus controls.

In general, this manuscript is well-written and logically designed. The technical approach is sound and systematic. However, there are some fairly serious issues that reduce enthusiasm, as listed below:

1. The title strongly suggests that this manuscript is primarily a “methods paper,” as there is really no description of why acetylation could be important in disease? While this is okay, the authors might consider revising to better define a “science message” if one is available.

Reply:

We appreciate this comment from the reviewer. The primary outcome of this study is a methodological development as no previous study has examined the lysine acetylome of human gut microbiome. A secondary outcome of this methodological advancement is to reveal the widespread distribution of lysine acetylation in gut microbial species as well as various metabolic pathways, including short-chain fatty acid (SCFA) production through anaerobic fermentation. We regret that we didn't deliver such important information. Therefore, we have expanded Results sections 3 and 4 (adding two new figures, as Fig. 2 and 3). Accordingly, we have changed the title to “*Widespread protein lysine acetylation in gut microbiome and its alterations in patients with Crohn's disease*”.

Although we have demonstrated the alterations of gut microbial lysine acetylome in pediatric CD patients compared to control, we don't want to over-interpret the results given the small patient cohort in this study. We have indicated this limitation of the current study in the Discussion section of the revised manuscript in Page 14 Lines 440-442, as follows:

“The current study was limited by the number of CD patients, however, the findings provide valuable information for designing further studies to understand the functionality of the microbiome in CD”

2. Abstract (line 25) and Introduction - of all the possible PTMs, why focus on acetylation? The rationale is very weak and vague, which diminishes the overall impact of the study. Are there expectations that more or less acetylation is associated with disease somehow?

Reply:

To provide better rationale to study lysine acetylation in the gut microbiome, we have added more background/literatures in the Introduction section. Briefly, we highlighted that protein acetylation was among the most abundant PTMs in bacterial species. We also highlighted that lysine acetylation has been reported to extensively regulate central metabolic pathways, including acetate metabolism, in bacterial species, such as *E. coli* and *Salmonella*. On the other hand, the metabolic intermediate of acetate metabolism, namely acetyl-phosphate and acetyl-CoA, can greatly shape the overall profiles of protein acetylation in bacteria. These are the reasons why the study of acetylation is of particular importance in prokaryotes, and for microbial communities such as the gut microbiota.

Short-chain fatty acid (SCFA) metabolism, including acetate metabolism, is among the most important metabolic functions of gut microbiota and has been widely reported to be associated with diseases, such as Crohn's disease. The study of protein acetylation of gut microbiomes, in particular in SCFA metabolism pathways, in patients of Crohn's disease may therefore aid in better understanding the role of microbiome in Crohn's disease. We have elaborated on these important aspects to provide better rationale in the revised manuscript, as follows (Pages 2-3 Lines 52-72):

“Compared with other PTMs that are commonly implicated in regulation of metabolic processes, such as phosphorylation, acetylation demonstrated higher levels in microorganisms¹⁰. In bacteria, up to 40% of proteins can be acetylated¹¹, due to the presence of both enzymatic and non-enzymatic acetylation mechanisms^{12, 13, 14, 15}.

Protein **Kac** has been characterized in several single bacterial species, including *Escherichia coli*^{13, 16, 17, 18}, *Bacillus subtilis*¹⁹, *Salmonella enterica*⁸ and *Mycobacterium tuberculosis*²⁰, and widely implicated in various microbial processes including chemotaxis²¹, nutrient metabolism¹⁸, stress response¹⁸ and virulence²². In *E. coli*, the enzymatic activities in acetate metabolism were regulated by acetylation¹⁸. On the other hand, metabolic intermediates of acetate metabolism, such as acetyl phosphate and acetyl-CoA, can non-enzymatically acetylate metabolic enzymes or provide acetyl donor for enzymatic lysine acetylation. Therefore, microorganisms may evolve elegant mechanisms in regulating cellular metabolism through acetylation¹².

One of the most important metabolic functions of the gut microbiome is fermentation of indigestible dietary fibers to generate short-chain fatty acids (SCFAs)²³. SCFAs can nourish the intestinal cells, maintain the acidic intestinal environment and thereby protecting the intestinal barrier function²⁴. Accumulating evidence suggests that intestinal SCFAs and SCFA-producing bacteria at least partially mediate the complex host-microbiome interactions that underlie the development of many diseases, such as CD^{3, 25}. Given the potential role of Kac in regulating SCFA metabolism^{18, 26}, the study of Kac in human gut microbiome may aid in better understanding the role of gut microbiome in CD.”

In addition, we have also added a brief description on lysine acetylation in the Abstract, as follows. Due to the 150-word length limit of abstract, we didn't expand this in the abstract.

“Lysine acetylation (Kac), an abundant post-translational modification (PTM) in prokaryotes, regulates various microbial metabolic pathways. However, no studies have examined protein Kac at the microbiome level, and it remains unknown whether Kac level is altered in patient microbiomes.” (Page 1, Line 21-24)

3. Line 98 – the authors should discuss the rationale for using the integrated gut microbial gene catalog vs. a matched assembled metagenome of the exact samples should be discussed and defended. How do the authors know that the gene catalog adequately reflects the exact nature and identify of the microbiota in their patients, especially the diseased cohort?

Reply:

We thank the reviewer for this comment. The matched metagenome database has been used for metaproteomic studies for decades, however one disadvantage of this approach is that it adds additional cost which may greatly hamper the application of metaproteomics in microbiome studies. As an alternative approach, we and others have proposed to use the gene catalog database when no metagenomic data set is available (refs: Zhang, Xu, et al. "MetaPro-IQ: a universal metaproteomic approach to studying human and mouse gut microbiota." *Microbiome* 4.1 (2016): 31; Mills, Robert H., et al. "Evaluating metagenomic prediction of the metaproteome in a 4.5-year study of a patient with Crohn's disease." *mSystems* 4.1 (2019): e00337-18.). This approach not only lowers the cost/barrier of metaproteomics studies but also makes the results comparable across different data sets as the same reference database is used. In addition, both our previous study and the study by Mills *et al.* also

demonstrated that the IGC database performed similarly to the matched metagenome database for the sensitivity and specificity of peptide identification.

We agree with the reviewer that the patient microbiomes, in particular those from MLI aspirate samples used in the current study, might have different taxonomic and gene compositions. Therefore, in this study, we have used an “augmented IGC database” for MLI data set, which includes the original IGC database, metagenome database of representative MLI aspirate samples, viral database as well as protein sequences from fungi (more details were described in Page 17 Lines 544-550 of the revised manuscript).

To provide more information and clarity, we have added the below descriptions to the Results and Methods sections in the revised manuscript:

“We, and others, have previously shown that the Integrated Gene Catalog (IGC) database³⁰ performed similarly to the matched metagenome database for metaproteomic identification^{31,32}. Therefore, in this study, we used the IGC database for the identification of both metaproteomic and lysine acetylomic data sets.” (Page 4 Lines 97-100)

“In this study, we used the human fecal microbial Integrated Gene Catalog (IGC) database for adult stool samples and the protein database that was generated in our previous metaproteomic studies of pediatric MLI aspirate samples³, which we termed the IGC+ database in the current study, for MLI aspirate samples. Briefly, the IGC+ database consists of protein sequences from human fecal microbial Integrated Gene Catalog (IGC) database³⁰, NCBI viral proteins, predicted protein sequences from shotgun metagenomic sequencing of MLI aspirate samples, and representative fungal species (details in³)” (Page.17 Lines 544-550)

4. Line 114 – the observation of only 6 Kac peptides common in the enrichment compared to the 117 observed in the metaproteome is confusing and concerning. This issue is revisited on line 363, where the authors suggest that Kac peptides in the unenriched samples are likely false identifications. This is confusing and raises serious concerns with respect to false positive control in the other samples, and whether many of those might be false as well?

Reply:

We regret our lack of clarity for the discussion on potential false identification of Kac peptides in unenriched aliquots. In shotgun proteomics or metaproteomics, a target-decoy database search strategy

and FDR filtering (1% threshold) were usually used to confidently identify peptides from MS spectra (ref: Elias, Joshua E., and Steven P. Gygi. "Target-decoy search strategy for increased confidence in large-scale protein identifications by mass spectrometry." *Nature methods* 4.3 (2007): 207-214.). In the current study, >47,000 peptides were identified from the unenriched aliquot and only 117 (0.2%) were Kac peptides, which is lower than the 1% FDR threshold. To examine whether those Kac peptides were of less quality, we checked the distribution of peptide-spectrum match (PSM) scores and PEP (posterior error probability of the identification), two indexes of the quality of each PSM. As shown in the figure below (Supplementary Fig. 10 in the revised manuscript), the 117 Kac peptides identified in metaproteome data set showed obviously lower scores and higher PEP than non-Kac peptides in the same aliquot. Same distribution of both PSM score and PEP was obtained when comparing the identified Kac peptides in acetylomic aliquot with those non-Kac peptides in metaproteomic aliquot, suggesting high confidence of identified Kac peptides in the enriched aliquot of this study.

To provide better clarity, we have added more discussion and included the figure below as supplementary Fig. 10 in the revised manuscript, as follows:

“It is worth noting that only six out of the 117 Kac peptides that were identified in unenriched samples overlapped with those identified in the enriched samples. **Given that only 0.2% of the peptides identified in unenriched aliquot were Kac peptides (less than the FDR threshold of 1% for target-decoy database search) and their lower peptide-spectrum match (PSM) scores (Supplementary Fig. 10),** the Kac peptides identified from the unenriched aliquot were **potentially** false identifications. **This finding further** suggests that an enrichment step during sample preparation is necessary to **deeply and reliably identify protein lysine acetylation** in the microbiome.” (Page 12 Lines 359-366)

Supplementary Fig. 10 Peptide-spectrum matching score (a) and PEP (b) distribution of identified acetylated peptides in both enriched and unenriched aliquots.

5. Line 117 – it seems a bit surprising that 80% of all protein groups have acetylation – was this expected? In light of the point above, this raises issue about false positives here again?

Reply:

We regret this confusing statement in the manuscript. While the data itself did show that 80% of the protein groups identified in the current study have lysine acetylation sites identified, it doesn't necessarily mean the same proportion of the proteins present in the samples can be acetylated. In this study, the same mass spectrometry parameters, including running time (2 hours), were used for both metaproteomic and lysine acetylotomic aliquots. However, the peptides and their equivalent protein amount injected to the MS were not comparable. For metaproteomics, peptides equivalent to 250 ng of proteins were injected to MS, however the Kac enriched peptides injected for MS analysis were equivalent to 2 mg of starting proteins, which is nearly 10,000-fold higher than metaproteomics (Page 17 Lines 520-524). This difference may lead to the consequence that some low abundant proteins (with Kac) were only identified in the enriched aliquots but not in unenriched aliquots (~15% as indicated in Page 4 Line 115). We expected that 20-40%

of proteins in microbiome that can be acetylated based on published data, but the current experimental setup cannot provide further evidence for such expectation.

To clarify, we have removed the confusing statement and the current descriptions are as follows:

“**Evaluating** the overlap of identified Kac proteins with proteins identified in unenriched samples, this study identified 25,144 protein groups and 3814 (15%) were only inferred from Kac modified peptides (Fig. 1d), suggesting an efficient enrichment of low abundant Kac proteins/peptides using the current enrichment approach.” (Page 4 Lines 114-117)

6. Line 156 – Do the authors suspect that the highest number of acetylation sites simply reflect the fact that Firmicutes and Bacteroidetes are simply the most abundant? In that case, acetylation might not be related to microbiome condition.

Reply:

We thank the reviewer for this comment. The reviewer is correct that the highest number of identified acetylation sites from Firmicutes and Bacteroidetes might be due to the high abundances of these two phyla in human gut microbiota, as the mass spectrometer tends to detect high abundant peptides. While the exact numbers of Kac peptides identified were of no important biological meaning, the most interesting observation in this study is that different microorganisms have different levels of lysine acetylation and certain microbial species in disease patients also showed different acetylation levels.

Briefly, we found that the genus *Fusicatenibacter* (mainly species *F. saccharivorans*) had the highest acetylome-to-metaproteome ratio (a median of 14.24), while *Homo sapiens* (human) had the lowest ratio (a median of 0.07) among those quantified taxa (Fig. 2b; as shown below). This indicates that the protein acetylation level is much higher in prokaryotes than in humans from the microbiome samples. Firmicutes was the only phylum that showed significantly higher percentages in the lysine acetylome than that in metaproteome, while the phyla Actinobacteria and Proteobacteria showed significantly lower percentages in acetylome (Fig. 2b). We also calculated the Firmicutes-to-Bacteroidetes (F/B) ratios based on the sum intensities of their distinctive peptides yielding an average of 6.35 in lysine acetylome, which was significantly higher than that of metaproteome (an average of 4.90; $P = 0.04$). These findings suggest that Firmicutes had relatively higher protein acetylation levels than Bacteroidetes.

Accordingly, we have added these new results to the revised manuscript in Page 6 Lines 156-170. In addition, we have also added discussions on the potential explanation of why Firmicutes may have higher levels of lysine acetylation in the Discussion section in Page 12 Lines 379-390, as follows:

“To explore whether Kac levels differed by taxa, the ratio of relative abundance in lysine acetylome to that in metaproteome was calculated for each taxon (Fig. 2b). The genus *Fusicatenibacter* (mainly species *F. saccharivorans*) had the highest acetylome-to-metaproteome ratio (a median of 14.24), while *Homo sapiens* (human) had the lowest ratio (a median of 0.07) (Fig. 2b). This indicates that the protein acetylation level is much higher in Prokaryotes than human proteins in the microbiome samples. Firmicutes was the only phylum that showed significantly higher percentage in lysine acetylotomic aliquot than that in metaproteomic aliquot ($p = 0.03$, paired Wilcoxon signed rank test), while Actinobacteria and Proteobacteria showed significantly lower percentages in acetylotomic aliquot (Fig. 2b). No significant difference was observed for Bacteroidetes despite its lower acetylome-to-metaproteome ratio (Fig. 2b). We calculated the Firmicutes-to-Bacteroidetes (F/B) ratios based on the intensities of their distinctive peptides yielding an average of 6.35 in lysine acetylome, which was significantly higher than that of metaproteome (an average of 4.90; $P = 0.04$, paired Wilcoxon signed rank test), further indicating higher protein acetylation levels in Firmicutes.” (Page 6 Lines 156-170)

“Acetyl-phosphate is a key metabolic intermediate in acetate metabolism, abundantly present in SCFA-producers in gut microbiota. Firmicutes is one of the most abundant bacterial phyla and major SCFA-producing bacteria, which plays important roles in human health at least in part through generating SCFAs and harvesting energy from indigestible dietary fibres^{45, 46}. Accordingly, nearly half of the identified Kac peptides in both adult stool and pediatric MLI aspirate samples were derived from Firmicutes. This is also in agreement with the observations that most acetyl-phosphate generating enzymes identified in this study are derived from Firmicutes and the latter had higher lysine acetylome-to-metaproteome ratios than other bacterial phyla. Interestingly, we found that Kac is a common PTM event for almost all the important enzymes in SCFA metabolism in gut microbiome, which may be due to non-enzymatically acetylation by the excessive acetyl-phosphate within the cellular compartment.” (Page 12 Lines 380-391)

Fig. 2 (b) Lysine acetylome-to-metaproteome ratios of quantified phyla and genera in human gut microbiome. The ratios were log₂-transformed for plotting. High indicates higher lysine acetylation levels, and low indicates lower lysine acetylation levels.

Another example we have added to the revised manuscript is that *Bacilli* was significantly decreased in CD patients compared to controls in metaproteome data set (figure below; Fig. 5c in the revised manuscript), however, it was significantly increased in CD in lysine acetylome data set. The ratio of acetylome-based abundance to metaproteome-based abundance of *Bacilli* was therefore significantly increased in CD compared to controls ($P < 0.0001$; Fig. 5d). These observations highlight the additional information provided by lysine acetylome and suggest that the relative levels of lysine acetylation are related to the microbial origin and the conditions of microbiome.

Fig. 5 (c) Percentage of *Bacilli* in metaproteome and lysine acetyome data sets when comparing CD with Control subjects; (d) Acetyome-to-metaproteome ratios of *Bacilli* in pediatric CD and control subjects.

7. Line 216 – is there a basis to expect protein acetylation to be important in CD patients? Why this particular focus group? What will the “differences” mean? Line 383 provides some speculation, but even this linkage is quite vague and suspect.

Reply:

We would like to thank the reviewer for this comment. One of the important reasons why we selected Crohn’s disease as an example for demonstrating the applicability of the developed approach is that this disease was repeatedly reported to be associated with altered intestinal SCFA levels or abundances of SCFA-producers in gut microbiota. Acetate, propionate and butyrate are three major SCFAs produced by human gut microbiota through anaerobic microbial fermentation. Interestingly, key metabolic intermediates of SCFA production, namely acetyl-phosphate and acetyl-CoA, were considered as major contributors of lysine acetylation in prokaryotes through either enzymatic and non-enzymatic reactions. These findings indicate a potential link of lysine acetylation with the SCFA metabolism in gut microbiota.

Coincidentally, in several bacterial species, lysine acetylation has also been reported to regulate central metabolic pathways, including acetate metabolism. Therefore, we proposed that gut microbial lysine acetylation may play an important role in diseases that are known to be associated with altered SCFA-producers, such as Crohn’s disease. Accordingly, we have modified the Introduction section to provide more rationale, as follows:

“One of the most important metabolic functions of the gut microbiome is fermentation of indigestible dietary fibers to generate short-chain fatty acids (SCFAs)²³. SCFAs can nourish the intestinal cells, maintain the acidic intestinal environment and thereby protecting the intestinal barrier function²⁴. Accumulating evidence suggests that intestinal SCFAs and SCFA-producing bacteria at least partially mediate the complex host-microbiome interactions that underlie the development of many diseases, such as CD^{3, 25}. Given the potential role of Kac in regulating SCFA metabolism^{18, 26}, the study of Kac in human gut microbiome may aid in better understanding the role of gut microbiome in CD.” (Page 3 Lines 65-72)

In this study, we didn’t discuss much on the biological meanings of differences between CD and control, given the small sample size of the cohort, to avoid over-interpretation of the data. Instead, we focused more on the biological importance of the taxonomic origin, functional distribution as well as structural characteristics of identified Kac proteins. We have further elaborated how the current data set/findings support the relationship between lysine acetylation and gut microbial SCFA metabolism in the Discussion section (details are in our reply to the below reviewer comment). We have also clearly indicated that, in the Discussion section, further studies targeting specific Kac sites in larger cohorts would be needed to make more confident biological conclusions.

“The current study was limited by the number of CD patients, however, the findings provide valuable information for designing further studies to understand the functionality of the microbiome in CD.” (Page 14 Lines 440-442)

8. The general outcome of this work is a broad and descriptive summary of protein acetylation, but delivers minimal biological value about what these differences mean?

Reply:

We appreciate the reviewer’s comment and regret our failure to deliver more biological messages in our original submission. This is the first study to comprehensively examine the gut microbial lysine

acetylome, an important step for the study of human microbiome functionality. The secondary outcome of this study is to demonstrate the widespread distribution of lysine acetylation in gut microbial metabolic pathways, in particular SCFA metabolism and to reveal the taxon-specific patterns of lysine acetylation in the microbiome of Crohn's disease patients. To provide more biological meanings of the identified Kac proteins, in the revised manuscript we have re-designed the Results section and dramatically expanded section 3 (*Phylogenetic variations of protein Kac levels in microbiome*) and 4 (*Widespread protein Kac in gut microbial metabolic pathways*).

Briefly, in the 3rd section (Pages 5-7 Lines 150-188), we calculated the acetylome-to-metaproteome ratios for all quantified genera/phyla, which highlighted the high level of lysine acetylation in Firmicutes compared to other phyla. We then tried to provide evidence for the relationship between acetyl-phosphate producing enzymes and lysine acetylation levels in the microbiome. We found significant correlations between the abundances of acetate kinase (ACK) and phosphate acetyltransferase (PTA) in metaproteome aliquots and the total abundance of all Kac proteins identified in acetylotomic aliquots, and the identified ACK or PTA proteins were mainly distributed in species from Firmicutes. In the 4th section (Pages 7-8 Lines 189-246), following functional profiling, we performed metabolic module construction using all identified Kac proteins, which yielded 51 complete metabolic modules, including the key pathways for generating short-chain fatty acids through anaerobic microbial fermentation. We then performed structural analysis of identified lysine acetylated proteins to demonstrate whether the identified acetylation sites are important for regulating enzymatic activities.

In addition, we have also expanded the discussion on potential link between SCFA metabolism and lysine acetylation in the revised manuscript in Pages 12-13 Lines 367-408, as follows:

“In bacteria, non-enzymatic acetylation by acetyl-phosphate has been considered as a major contributor for protein acetylation^{13, 44}. In enzymatic acetylation mechanism, a catalytic glutamate (E) residue in the enzyme is required to deprotonate the epsilon-amino group of the target lysine¹². Similarly, an internal acidic amino acid, such as E or D, near the target lysine is required to deprotonate the epsilon-amino group in a non-enzymatic mechanism¹². Accordingly, we found that the -1 position of microbiome Kac site was significantly enriched by E and D (top 1 and 2, respectively; Fig. 1f), suggesting that non-enzymatic acetylation mechanism is predominantly present in the gut microbiome. In addition, the relative abundance of enzymes for the generation of acetyl-phosphate from acetyl-CoA significantly correlated with the overall Kac levels in microbiome samples, while ACAT (converts acetyl-CoA to acetoacetyl-CoA for the production of butyrate) negatively correlated with the overall Kac levels. These findings,

for the first time, provide evidence for a non-enzymatic protein acetylation mechanism in prokaryotes at the microbiome level.

Acetyl-phosphate is a key metabolic intermediate in acetate metabolism, abundantly present in SCFA-producers in gut microbiota. Firmicutes is one of the most abundant bacterial phyla and major SCFA-producing bacteria, which plays important roles in human health at least in part through generating SCFAs and harvesting energy from indigestible dietary fibres^{45, 46}. Accordingly, nearly half of the identified Kac peptides in both adult stool and pediatric MLI aspirate samples were derived from Firmicutes. This is also in agreement with the observations that most acetyl-phosphate generating enzymes identified in this study are derived from Firmicutes and the latter had higher lysine acetylome-to-metaproteome ratios than other bacterial phyla. Interestingly, we found that Kac is a common PTM event for almost all the important enzymes in SCFA metabolism in gut microbiome, which may be due to non-enzymatically acetylation by the excessive acetyl-phosphate within the cellular compartment. Castano-Cerezo *et al.* previously reported that many proteins involved in acetate metabolism, including ACS which converts acetate to acetyl-CoA, are acetylated proteins and their activities are also regulated by lysine acetylation¹⁸. In *Salmonella*, Wang *et al.* demonstrated that enzymes in central metabolic pathways were extensively acetylated and protein acetylation regulated the direction of carbohydrate metabolic flux in response to environmental changes⁸. In this study, the structural analysis of PCK, one of the most abundant Kac enzymes, also suggested that acetylation might be involved in regulating the direction of SCFA metabolism. We found that the catalytically essential structure α -loop of PCK, which regulates enzyme conformation^{35, 36}, was among the most abundantly acetylated proteins in the microbiome. The acetylation of K473 in rat PCK α -loop, which shares highly similar secondary structure to that of bacterial PCK (Fig. 3d), has been shown to significantly increase the efficiency of conversion from phosphoenolpyruvate (PEP) into oxaloacetate, while decrease the efficiency of gluconeogenic reaction (oxaloacetate to PEP)⁴⁷. This suggests that the identified Kac site on gut microbial PCK might be involved in accelerating the metabolic flow of PEP to oxaloacetate and thereby succinate/propionate (Fig. 3b). Taken together, these findings suggest that gut microbial Kac might be an important mechanism regulating the SCFA metabolism and influences the complex host-microbiome interactions in diseases.”

9. Line 417 – One of the most concerning and unsettling aspects of this study is the sample cohort. The number of patients are low but not unacceptable for such a study. The most serious concern is that protein acetylation is known to be dynamic and thus temporal aspects are critical here. The use of only one sample per patient makes this study without context – how do the authors know that collection of samples from the same patients at another time-point wouldn't produce dramatically different results. This could be especially true in the diseased patients. This study would have been more meaningful if the authors had conducted multiple time-point measurements on a more limited number of the samples.

Reply:

We thank the reviewer very much for this important comment. To address this concern, we evaluated the dynamic characteristics of protein lysine acetylation in microbiomes using two newly generated data sets.

The first data set included 7 stool samples collected from two healthy adults (V51 and V52) at four different days within 1 week, i.e., day 1 (Wednesday), day 2 (Thursday), day 3 (Friday, *missing V51*) and day 7 (Tuesday). The second data set includes 8 MLI aspirate samples collected from three pediatric Crohn's disease patients prior to and after therapeutic treatments (up to 46 months on therapeutic treatments). Details of the patient cohort, including the disease severity, inflammatory status, time since diagnosis and medications applied, are shown in the table below (Supplementary Table 6 in the revised manuscript). In total, 15 microbiome samples were collected and all samples were processed for both metaproteomic and lysine acetylomic analysis using the same experimental and bioinformatics workflows described in the manuscript.

Supplementary Table 6 Sample information of pre- and post-treatment cohort

[Redacted]

In total, 6,665 protein groups and 6,286 Kac sites were identified for stool samples collected from two healthy adults, and 10,085 protein groups and 9,225 Kac sites were identified for the MLI aspirate samples collected from three pediatric CD patients. We first evaluated the overall profiles of metaproteome and lysine acetylome using Principal component analysis (PCA). As shown in the figure below (Fig. R1), the samples collected from the same individual clustered closely together for metaproteome as well as lysine acetylome in both healthy adults (Fig. R1a, b) and pediatric CD patients (Fig. R1d, e). Given that the abundances of lysine acetylated proteins might only reflect the abundances of total proteins, we also calculated the ratios between the abundances of lysine acetylated proteins and their corresponding protein abundances in metaproteomic aliquot, a proxy of the relative lysine acetylation level. As shown in Fig. R1c and f, PCA analysis of acetylome-to-metaproteome ratios also showed that the samples collected from the same individual clustered together. These findings suggest that the lysine acetylome of individual's microbiota is relatively stable over time when compared to inter-individual variations. This phenomenon is also in agreement with and/or due to the known high intra-individual stability of both functional- and strain-level compositions in human gut microbiota (Costea, Paul I., et al. "Subspecies in the global human gut microbiome." *Molecular systems biology* 13.12 (2017); Faith, Jeremiah J., et al. "The long-term stability of the human gut microbiota." *Science* 341.6141 (2013): 1237439.). Therefore, single time-point measurement of lysine acetylome in this study is valid and the collection of samples at another time-point from the same patients is not likely to change the main conclusions of the current study.

Fig. R1 PCA analysis of metaproteome and lysine acetylome of adult stool (a-c) and pediatric MLI microbiota (d-f). PCA score plots of metaproteome (a, d), lysine acetylome (b, e), as well as their ratios (c, f) were shown. S00 indicates the samples collected prior to treatment; S01 and S01 indicate the first and second post-treatment samples, respectively. Samples from the same patient were in the same color and connected with lines.

To further demonstrate whether the findings in the manuscript were meaningful, we evaluated the changing trend of taxa and representative Kac sites during disease alleviation for each individual patient. In the case-control cohort in our original submission, we found that, among the five most abundant phyla, Actinobacteria, Bacteroidetes and Proteobacteria showed increased acetylome-to-metaproteome ratios in CD compared to control, and Firmicutes and Verrucomicrobia showed decreased acetylome-to-metaproteome ratios (Supplementary Fig. 5). Therefore, we first evaluated the changing trend of the acetylome-to-metaproteome ratios of the above five abundant phyla. Interestingly, we found that, in this small therapeutic cohort, the acetylome-to-metaproteome ratios of all five phyla showed a changing trend toward to that of control subjects (Supplementary Fig. 7). *Bacilli* and *Ruminococcus*, the two taxa that showed increased acetylome-to-metaproteome ratios in CD compared to control, also showed a trend of decreased acetylome-to-metaproteome ratios when the patients were in remission in the first post-treatment time point (Supplementary Fig. 7).

Supplementary Fig. 5 Lysine acetylome-to-metaproteome ratios of gut microbial phyla in CD and control patients. The bottom and top of the boxes are the first and third percentile, respectively. The middle line represents the median (50th percentile). Whiskers are drawn from the ends of the interquartile range (IQR) to the furthest observations within 1.5 times the IQR range. Outliers >1.5 times the IQR are indicated with black dots.

Supplementary Fig. 7 Lysine acetylome-to-metaproteome ratios of taxa quantified in pediatric CD patients before and after treatment. S00 indicates the samples collected prior to treatment; S01 and S01 indicate the first and second post-treatment samples, respectively.

We also evaluated the Kac levels of calprotectin proteins, namely S100A8 and S100A9, which were among the most significantly changed host proteins in their Kac levels identified in case-control cohort in our original submission. In this small therapeutic cohort, we totally quantified 5 Kac sites for S100A8 and 6 Kac sites for S100A9. As shown in the figure below (Supplementary Fig. 9), the overall Kac levels of both S100A8 and S100A9 showed decreasing trend after treatment. Among the Kac sites identified in this study, a general trend of decreasing was observed, in particular for those that have been identified as being significantly increased in CD compared to control, namely S100A8 K18, K35, S100A9 K4 and K38. Taken together, these results further validate that the alterations of lysine acetylome in CD compared to control subjects identified in our manuscript are meaningful.

Accordingly, we have added this part of results to the revised manuscript as a further validation of the case-control study in the Results section, as follows:

“In this study, we also analyzed the lysine acetylomes of time-series MLI aspirate samples collected from three additional pediatric CD patients who were undergoing disease alleviation following treatment (Supplementary Table 6). In total, 10,085 protein groups and 9,225 Kac sites were identified for metaproteomic and lysine acetylotomic aliquots, respectively. PCA analysis showed that samples collected from the same patients clustered together for both metaproteome and lysine acetylome, as well as their ratios, albeit collected up to 46 months apart and with disease alleviation (Supplementary Fig. 6). This result suggests that both the metaproteome and lysine acetylome of an individual’s microbiome are relatively stable over time. This is in agreement with previous metagenomic studies showing long-term stability of both functional- and strain-level compositions of gut microbiota^{38, 39}. Interestingly, we found that the acetylome-to-metaproteome ratios of abundant phyla, namely Actinobacteria, Bacteroidetes, Proteobacteria and Firmicutes, were all partially reverted during disease alleviation, in particular for the first post-treatment time point when all three patients were in remission (Supplementary Fig. 7). *Bacilli* and *Ruminococcus*, also showed a trend of decreased acetylome-to-metaproteome ratios when the patients were in remission in the first post-treatment time point (Supplementary Fig. 7).” (Page 10 Lines 295-310)

“Evaluating the changes of Kac sites of S100A8 and S100A9 following disease alleviation in the treatment cohort, we found that the overall Kac levels of both S100A8 and S100A9 showed decreasing trend after treatment (Supplementary Fig. 9). Among

the Kac sites identified in this therapeutic cohort, a general decreasing trend was observed, in particular for those that have been identified as being significantly increased in CD compared to control, such as S100A8 K18, K35, S100A9 K4 and K38 (Supplementary Fig. 9). These findings further validate the alterations of lysine acetylome identified when comparing CD with controls.” (Page 11 Lines 334-341)

Supplementary Fig. 9 Kac site-to-protein ratios of fecal S100A8 and S100A9 proteins in pediatric CD patients before and after treatment. S00 indicates the samples collected prior to treatment; S01 and S01 indicate the first and second post-treatment samples, respectively. To calculate the “Combined” Kac site-to-protein ratio, all quantified Kac sites for that protein were summed and divided by the LFQ intensity of proteins in metaproteome.

Reviewer #3 (Remarks to the Author):

In this manuscript, Zhang and colleagues have performed a characterization of the acetylation sites present in metaproteomic samples, including changes occurring in IBD patients. They have identified on the order of 30,000 acetylation sites in bacterial proteins, which they have mapped to bacterial taxonomic diversity and protein functional annotations. As expected, most acetylation maps to proteins that relate to translation and metabolism. Interestingly, by performing also a separate proteomic analysis the authors could determinate the ratio of acetylated vs. non-modified peptides for different taxonomic groups. Finally, they analysed a small number of IBD patients and controls and performed a proof-of-principle analysis of how acetylation changes in the microbial samples of these patients. I think the number of samples is too low but serves as a useful test case application. The large number of acetylation sites identified in the study serves as potential exciting resource to study protein regulation by acetylation. However, the authors don't really make an effort of exploring this resource. The taxonomic differences in ratio of proteome acetylation is an interesting aspect that is mentioned mostly in discussion but it seems to be the most interesting result of the manuscript and could be better analysed. As it currently stands, the manuscript is mostly a quick description of acetylation patterns and does not go very far in novel discoveries.

Reply:

We thank you very much for these insightful comments and agree that there was a lack of depth in exploring this large data set we generated. We greatly appreciate the reviewer for guiding us on how to dig into the data as described in the comments below. We have thoroughly revised our manuscript according to the reviewer's suggestions. Please find below detailed explanations of all added data analysis and modifications, which we believe greatly improved the manuscript.

Major concerns

The authors have identified what it is likely to be the largest collection of microbial acetylation sites to date. However, very little is done with this resource beyond a brief description of the taxonomic diversity and biological processes connected with these peptides. Understanding how most of the acetylation regulates protein activity is an important open question and this resource could be a fantastic opportunity to study this problem. The authors should provide some examples of how this could be achieved to motivate others to explore this great resource they have generated. For acetylated sites that match proteins

that exist across many bacterial genomes the authors could align the protein sequences and map the acetylation sites to the alignment in order to study regions that are often acetylated across a large number of proteomes/species. For a number of these, there will be structural models that can serve as homology templates to study the structural impact of mapped acetylation sites. Since many of these will be in metabolic enzymes there should be many enzymes that are present in many different species and a number of structural models available to represent these enzymes. For example, the authors could ask if acetylation sites are often near catalytic residues for example.

Reply:

We thank the reviewer for these constructive comments. Accordingly, we have extensively expanded this part of results according to the suggestions by the reviewer (now presented as an individual section in Results section 4 “*Widespread protein Kac in gut microbial metabolic pathways*”, as well as a new Fig. 3). Briefly, we first performed metabolic module construction using all identified Kac proteins, which yielded 51 complete metabolic modules, including the key pathways for generating short-chain fatty acids through anaerobic microbial fermentation (detailed pathways were elaborated in Fig. 3b). Then, we evaluated the most abundant Kac peptides identified in this study and revealed that most of these peptides were shared by taxa across the kingdom Bacteria or by specific taxa that are known to produce SCFAs in human gut. We then took phosphoenolpyruvate carboxykinase (ATP-dependent) (PCK) as an example to demonstrate whether the identified acetylation sites are important for regulating enzymatic activities from the protein structure point-of-view. The added results and figures are as follows (Pages 7-8):

“Kyoto Encyclopedia of Genes and Genomes (KEGG) annotation showed that 11,536 out of the 15,053 Kac proteins (76.6%) were mapped to 1354 KEGG orthologies (KOs) and 224 pathways. Among the 1354 KOs, 994 were enzymes and 626 were mapped to metabolic pathways. Fifty-one complete metabolic modules were constructed using Kac proteins, including glycolysis, citrate cycle, gluconeogenesis, pyruvate oxidation and dissimilatory sulfate reduction (Supplementary Table 3). Intestinal microbiota is known to process complex carbohydrates, such as indigestible dietary fibre, to generate SCFAs that maintain the homeostasis of the intestinal microenvironment²³. We found that carbohydrate metabolism was the most widely acetylated metabolic pathway in microbiome, in particular pyruvate metabolism (49 KOs), amino sugar and nucleotide sugar metabolism (46 KOs), glycolysis/gluconeogenesis (41 KOs), fructose and mannose metabolism (39 KOs), butanoate metabolism (36 KOs), propanoate metabolism (35 KOs), and starch and sucrose metabolism (35 KOs) (Supplementary Data 5). In addition, we established complete anaerobic fermentation pathways that

produce SCFAs using the identified Kac enzymes in this study (Fig. 3b), indicating a widespread protein acetylation of the enzymes involved in these important gut microbial metabolic pathways.

Examining the top 10 most abundant Kac peptides identified in this study, we found that nine were from bacteria and one from human chymotrypsin-like elastase family member 3A (Supplementary Table 4). Moreover, all the nine most abundant microbial Kac peptides were from enzymes involved in the SCFA production, including glyceraldehyde-3-phosphate dehydrogenase (GAPDH, 3 peptides), 3-phosphoglycerate kinase (PGK, 2 peptides), pyruvate: ferredoxin oxidoreductase (PFOR, 2 peptides), phosphoenolpyruvate carboxykinase (ATP-dependent) (PCK, 1 peptide), and 3-hydroxyacyl-CoA dehydrogenase (HADH, 1 peptide) (Fig. 3b). Six of the nine microbial Kac peptides had the lowest-common ancestor (LCA) of Bacteria or root (shared by all organisms), while the other three were unique to species belonging to *Clostridiales* (Supplementary Table 4), the major SCFA-producers in microbiota.

We then took PCK as an example to examine whether the identified Kac site is important for regulating enzymatic activity. PCK catalyzes the reversible conversion of phosphoenolpyruvate (PEP) into oxaloacetate (Fig. 3b). A blastp search against NCBI-nr database revealed high sequence similarity of PCKs across different bacterial species (>89% identity and >98% coverage for top 100 hits; >99% hits were from Firmicutes). Alignment of the identified PCK protein sequence in this study to known PCK proteins in Protein Data Bank (PDB) identified *Anaerobiospirillum succiniciproducens* PCK as the most similar one (full length protein sequence identity of 82% and E value of 1E-81) (Fig. 3c). There are three essential catalytic structural elements in PCK, namely P-loop, R-loop, and Ω -loop (Fig. 3d). P-loop and the R-loop are directly involved in catalysis and substrate binding, while Ω -loop act as a lid-gate by switching from a closed-active conformation to an open-inactive conformation^{35, 36}. *E. coli* PCK with the truncated or shortened Ω -loop has been reported to loss enzyme activity³⁷. In the crystal structure of *A. succiniciproducens* PCK with a closed-active conformation, the mapped site K384 at Ω -loop forms a salt-bridge with glutamine 389 (E389) which interacts with key catalytic residue arginine 60 (R60) located at R-loop (Fig. 3e). Therefore, the acetylation of K384 could interrupt these interactions and de-stabilize the active conformation of PCK. Further biochemical validation is still required, however, this finding suggests that the

exploration of the current data set helps the study of how protein acetylation may regulate gut microbial activity.”

Fig. 3 Functional characterization of identified Kac proteins. (a) COG category distribution of microbial Kac proteins. Significantly enriched categories are highlighted in orange. Significance was determined with a hypergeometric test using the unmodified microbial proteins identified in the metaproteomic samples as background. (b) SCFA-producing metabolic pathways constructed using the identified Kac proteins. Identified Kac enzymes and metabolites were indicated. (c) Sequence alignment of identified acetylated PCK (MH0173_GL0113524, Kac peptide GFTAKaLAGTER) with known PCKs in PDB database. Taxonomic origin and starting amino acid position are indicated in the left side. The consensus sequence is colored in blue gradient according to the percentage identity. *A. succinogenes*, *Actinobacillus succinogenes*; *E. coli*, *Escherichia coli*; *T. thermophiles*, *Thermus thermophiles*; *T. cruzi*, *Trypanosoma cruzi*. (d) GTP-dependent and ATP-dependent PCKs share the same catalytic structural elements. The structure of the catalytic pocket of the GTP-dependant rat PCK (colored in grey, PBD 3DT4) is superposed with *A. succiniciproducens* ATP-dependant PCK (colored in Cyan, PBD 1YTM).

Three catalytic elements: R-loop, P-loop, Ω -loop are highlighted with light-blue, light-red and light-yellow, respectively in rat PCK, and with bright-blue, bright-red and bright-yellow, respectively, in *A. succiniciproducens* PCK. The oxalate and ATP are indicated as sticks and colored by atom type. The Mg and Mn metals are indicated as green spheres. (e) Interaction among K384, E389, R60 and oxalate in *A. succiniciproducens* PCK. Protein structure was generated with PyMOL (<https://pymol.org/>).

In addition to PCK, we also examined the structural characteristics of the identified Kac sites of GAPDH, however, we only included this part of results in this response letter due to the length limit of the manuscript. GAPDH plays an essential role in glycolysis by catalyzing the oxidative phosphorylation of glyceraldehyde-3-phosphate (GA3P) to 1,3-bisphospho-glycerate (BPG) with co-factor NAD^+ . In both prokaryote and eukaryote, the enzyme primarily function as a homo-tetramer with each subunit containing a NAD^+ binding domain and a catalytic domain. The most abundant Kac peptide identified in this study located in the large S-shaped loop, called “S-loop”, in the catalytic domain of GAPDH. Sequence alignment suggests that the S-loop is conserved in both prokaryote and eukaryote (Fig. R2a). As an important structural feature of active site, S-loop contributes to the tetramer formation, NAD^+ binding, as well as allosteric activation (Fig. R2b) (Biesecker, G., *et al.* "Sequence and structure of D-glyceraldehyde 3-phosphate dehydrogenase from *Bacillus stearothermophilus*." *Nature* 266.5600 (1977): 328-333; Yun, Mikyung, *et al.* "Structural analysis of glyceraldehyde 3-phosphate dehydrogenase from *Escherichia coli*: direct evidence of substrate binding and cofactor-induced conformational changes." *Biochemistry* 39.35 (2000): 10702-10710.). Therefore, the post-translational modifications on S-loop may reduce GAPDH enzyme activity by dissociating the active homo-tetramer complex, interfering the NAD^+ binding, or blocking dynamically conformational change for activation. Accordingly, in *Toxoplasma gondii*, the phosphorylation of Serine (S) located in S-loop has been reported to block the enzyme activity of GAPDH (Dubey, Rashmi, *et al.* "Membrane skeletal association and post-translational allosteric regulation of *Toxoplasma gondii* GAPDH1." *Molecular microbiology* 103.4 (2017): 618-634.). There is no published data on the effects of lysine acetylation of S-loop on the enzymatic activity of GAPDH, the finding from this study suggests the presence of novel regulation mechanism of GAPDH activity through the S-loop lysine acetylation.

Fig. R2 Structural characterization of identified Kac site on GAPDH. (a) Sequence alignment of the S-loop. The acetylated peptide was assigned to *Bacteroides fluxus* GAPDH and the S-loop sequence was aligned with the other GAPDHs with known crystal structures including *Escherichia coli* (PBD 1U8F), *Toxoplasma gondii* (PBD 3STH), and human (PBD 1UBF). The acetylation site in the peptide, as well as the known phosphorylation site are indicated. (b) Surface view of the tetramer GAPDH (PBD 1UBF, 4). The subunits O, P, Q, and R are shown in green, light-blue, wheat, and pink respectively. The S-loop of each subunit forms the interface with other subunits. The S-loops of P, Q subunits are colored in marine and brick-red, respectively. The lysine residues involved in acetylation are highlighted with dark-blue and bright-red, respectively. NAD molecule is rendered as spheres and colored by atom type. (c) Overall structure of subunit P with the S-loop of subunit Q. Each subunit contains a N-terminal domain binding to NAD (cyan) and a C-terminal catalytic domain (light blue). The S-loop (dark blue) is located at C-terminal domain, which is highly flexible in NAD free structure and perform allosteric change during catalysis. The S-loop of subunit Q (brick-red) also contributes to bind NAD in the active site of neighboring subunit P. The lysine residues involved in acetylation are presented in sticks.

In my opinion, the most interesting aspect of the results presented were the potential differences in ratio of acetylated vs non-modified peptides along the taxonomic diversity studied. The authors speculated that this differences in ratios could be due to metabolic differences with some species being known producers of acetate. Acetate can be converted to acetyl-phosphate in some species which can in turn directly

chemically acetylate proteins (Weinert et al. Mol Cell 2013). The authors could try to further analyse this association using isolate genomes or very complete metagenomic assembled genomes. Can the authors find a relationship between the enzymes encoded in the genomes of some of the taxonomic groups and their ratios of acetylated vs non-modified peptides?

Reply:

We thank the reviewer for this insightful suggestion. Accordingly, we have expanded the taxonomic results to make a separate section in the Results as Section 3 “*Phylogenetic variations of protein Kac levels in microbiome*” in the revised manuscript. Briefly, in this section, we first summarised the taxonomic distributions of identified Kac peptides (figure below; as Fig. 2 in the revised manuscript). Then we calculated the acetylome-to-metaproteome ratios for all quantified genera/phyla (Fig. 2b), which highlighted the high level of lysine acetylation in Firmicutes, while low acetylation level on human proteins. Furthermore, we also tried to provide evidence for the relationship between acetyl-phosphate producing enzymes and lysine acetylation levels in the microbiome. We found significant correlations between the abundances of acetate kinase (ACK) and phosphate acetyltransferase (PTA) in metaproteome aliquots and the total abundance of all Kac proteins identified in acetylotomic aliquots (Fig. 2c-d). It is difficult to perform genome-resolved analysis in such as relatively low-depth data set. However, we extracted all the 87 identified ACK or PTA proteins in the metaproteomic aliquot and performed taxonomic assignment (Supplementary Table 1, as shown below), which showed that majority (62/87) of them belonged to Firmicutes. As shown in Fig. 2b, Firmicutes was the only phylum that showed significantly higher percentage in lysine acetylotomic aliquot than that in metaproteomic aliquot ($p = 0.03$, paired Wilcoxon signed rank test). In addition, some known butyrate- and/or acetate- producing bacterial species/genera, such as *Faecalibacterium*, *Lachnospira*, [*Eubacterium*] *rectale*, and *butyrate-producing bacterium* SS3/4, also have high number of ACK or PTA proteins in their proteomes. We have added these results to the revised manuscript in Pages 5-7, as follows:

“**Biodiversity analysis of the identified Kac peptides using Unipept³⁴ revealed that 28,321 peptides (80%) were assigned to the kingdom Bacteria and 24,785 peptides could be classified at phylum level (15,170 from Firmicutes, 7876 from Bacteroidetes and 1739 from other phyla; Fig. 2a and Supplementary Data 3). A high proportion of the Kac peptides were from bacteria belonging to four genera: *Prevotella*, *Faecalibacterium*, *Bacteroides*, and *Eubacterium* (Supplementary Data 3). To explore whether Kac levels differed by taxa, the ratio of relative abundance in lysine acetylome to that in metaproteome was calculated for each taxon (Fig. 2b). The genus *Fusicatenibacter* (mainly species *F. saccharivorans*) had the highest acetylome-to-metaproteome ratio (a**

median of 14.24), while *Homo sapiens* (human) had the lowest ratio (a median of 0.07) (Fig. 2b). This indicates that the protein acetylation level is much higher in Prokaryotes than human proteins in the microbiome samples. Firmicutes was the only phylum that showed significantly higher percentage in lysine acetylomic aliquot than that in metaproteomic aliquot ($p = 0.03$, paired Wilcoxon signed rank test), while Actinobacteria and Proteobacteria showed significantly lower percentages in acetylomic aliquot (Fig. 2b). No significant difference was observed for Bacteroidetes despite its lower acetylome-to-metaproteome ratio (Fig. 2b). We calculated the Firmicutes-to-Bacteroidetes (F/B) ratios based on the intensities of their distinctive peptides yielding an average of 6.35 in lysine acetylome, which was significantly higher than that of metaproteome (an average of 4.90; $P = 0.04$, paired Wilcoxon signed rank test), further indicating higher protein acetylation levels in Firmicutes.

Acetyl-phosphate, a metabolic intermediate of SCFA production, has been shown to be a critical contributor for protein lysine acetylation in prokaryotes¹³. To study the association of acetyl-phosphate with global Kac levels in the microbiome, we correlated the abundances of acetyl-phosphate-producing enzymes, namely acetate kinase (ACK) and phosphate acetyltransferase (PTA), in the metaproteomic aliquots with the total abundance of all Kac peptides identified in acetylomic aliquots. Significant correlations were obtained for both PTA ($R = 0.94$; $P = 0.02$; Fig. 2c) and ACK ($R = 0.77$; $P = 0.10$; Fig. 2d). No or negative correlation was observed when correlating acetyl-CoA synthase (ACS) and acetyl-CoA acetyltransferase (ACAT) with total abundance of Kac peptides (Supplementary Fig. 4). Taxonomic assignment of the 87 proteins annotated as ACK and PTA identified in this study showed that 62 proteins belonged to Firmicutes, and 25 belonged to other phyla including Bacteroidetes. *Bacteroides* (14 proteins), *Blautia* (13 proteins), *Faecalibacterium* (11 proteins), *Lachnospira* (8 proteins), *[Eubacterium] rectale* (6 proteins), butyrate-producing bacterium SS3/4 (5 proteins) and *Prevotella* (5 proteins) were the genera/species with the highest number of ACK or PTA proteins identified in the metaproteomic aliquots (Supplementary Table 1). These results are in agreement with the taxonomic distribution of Kac peptides identified in lysine acetylomic aliquots (Fig. 2a), and again suggest that Firmicutes had higher protein acetylation levels in microbiome.”

Fig. 2 Taxon-specific lysine acetylation patterns in human gut microbiome. (a) Sunburst plot of microbial taxa that were assigned using all of the identified Kac peptides. Sunburst plot was generated using Unipept (<https://unipept.ugent.be/>). (b) Lysine acetylome-to-metaproteome ratios of quantified phyla and genera in human gut microbiome. The ratios were log2-transformed for plotting. High indicates higher lysine acetylation levels, and low indicates lower lysine acetylation levels. (c-d) Correlations of overall Kac protein levels with the relative abundances of phosphotransacetylase (c) and acetate kinase (d) in metaproteome. Spearman's correlation R and P values are indicated.

Supplementary Table 1 Taxonomic assignment of 87 proteins annotated as acetate kinase and phosphotransacetylase

Phylum	Genus	Protein Count
Actinobacteria	Bifidobacterium	2
Actinobacteria	Collinsella	1
Bacteroidetes	Bacteroides	14
Bacteroidetes	Prevotella	5
Bacteroidetes	Odoribacter	1
Firmicutes	Blautia	13
Firmicutes	Faecalibacterium	11

Firmicutes	Lachnospira	8
Firmicutes	[Eubacterium] rectale	6
Firmicutes	butyrate-producing bacterium SS3/4	5
Firmicutes	Lacrimispora	4
Firmicutes	Enterocloster	3
Firmicutes	[Eubacterium] siraeum	2
Firmicutes	Anaerobutyricum	2
Firmicutes	Anaerostipes	2
Firmicutes	Butyrivibrio	2
Firmicutes	Dysosmobacter	1
	Hungateiclostridiaceae bacterium	
Firmicutes	KB18	1
Firmicutes	Roseburia	1
Firmicutes	Ruminococcus	1
Fusobacteria	Fusobacterium	1
Fusobacteria	Ilyobacter	1

In addition, we have also modified Introduction section and added a paragraph of Discussion in the revised manuscript to provide more context for acetyl-phosphate-mediated non-enzymatic lysine acetylation mechanism, as follows:

“In bacteria, up to 40% of proteins can be acetylated ¹¹, due to the presence of both enzymatic and non-enzymatic acetylation mechanisms ^{12,13,14,15}. Protein Kac has been characterized in several single bacterial species, including *Escherichia coli* ^{13, 16, 17, 18}, *Bacillus subtilis* ¹⁹, *Salmonella enterica* ⁸ and *Mycobacterium tuberculosis* ²⁰, and widely implicated in various microbial processes including chemotaxis ²¹, nutrient metabolism ¹⁸, stress response ¹⁸ and virulence ²². In *E. coli*, the enzymatic activities in acetate metabolism were regulated by acetylation ¹⁸. On the other hand, metabolic intermediates of acetate metabolism, such as acetyl phosphate and acetyl-CoA, can non-enzymatically acetylate metabolic enzymes or provide acetyl donor for enzymatic lysine acetylation. Therefore, microorganisms may evolve elegant mechanisms in regulating cellular metabolism through acetylation ¹².” (Introduction; Page 2 Lines 54-64)

“In bacteria, non-enzymatic acetylation by acetyl-phosphate has been considered as a major contributor for protein acetylation ^{13, 44}. In enzymatic acetylation mechanism, a catalytic glutamate (E) residue in the enzyme is required to deprotonate the epsilon-amino group of the target lysine ¹². Similarly, an internal acidic amino acid, such as E or

D, near the target lysine is required to deprotonate the epsilon-amino group in a non-enzymatic mechanism¹². Accordingly, we found that the -1 position of microbiome Kac site was significantly enriched by E and D (top 1 and 2, respectively; Fig. 1f), suggesting that non-enzymatic acetylation mechanism is predominantly present in the gut microbiome. In addition, the relative abundance of enzymes for the generation of acetyl-phosphate from acetyl-CoA significantly correlated with the overall Kac levels in microbiome samples, while ACAT (converts acetyl-CoA to acetoacetyl-CoA for the production of butyrate) negatively correlated with the overall Kac levels. These findings, for the first time, provide evidence for a non-enzymatic protein acetylation mechanism in prokaryotes at the microbiome level.

Acetyl-phosphate is a key metabolic intermediate in acetate metabolism, abundantly present in SCFA-producers in gut microbiota. Firmicutes is one of the most abundant bacterial phyla and major SCFA-producing bacteria, which plays important roles in human health at least in part through generating SCFAs and harvesting energy from indigestible dietary fibres^{45, 46}. Accordingly, nearly half of the identified Kac peptides in both adult stool and pediatric MLI aspirate samples were derived from Firmicutes. This is also in agreement with the observations that most acetyl-phosphate generating enzymes identified in this study are derived from Firmicutes and the latter had higher lysine acetylation-to-metaproteome ratios than other bacterial phyla. Interestingly, we found that Kac is a common PTM event for almost all the important enzymes in SCFA metabolism in gut microbiome, which may be due to non-enzymatically acetylation by the excessive acetyl-phosphate within the cellular compartment. Castano-Cerezo *et al.* previously reported that many proteins involved in acetate metabolism, including ACS which converts acetate to acetyl-CoA, are acetylated proteins and their activities are also regulated by lysine acetylation¹⁸. In *Salmonella*, Wang *et al.* demonstrated that enzymes in central metabolic pathways were extensively acetylated and protein acetylation regulated the direction of carbohydrate metabolic flux in response to environmental changes⁸. In this study, the structural analysis of PCK, one of the most abundant Kac enzymes, also suggested that acetylation might be involved in regulating the direction of SCFA metabolism. We found that the catalytically essential structure α -loop of PCK, which regulates enzyme conformation^{35, 36}, was among the most abundantly acetylated proteins in the microbiome. The acetylation of K473 in rat PCK α -loop, which shares highly similar secondary structure to that of bacterial PCK (Fig. 3d), has been shown to significantly increase the efficiency of conversion from phosphoenolpyruvate (PEP) into

oxaloacetate, while decrease the efficiency of gluconeogenic reaction (oxaloacetate to PEP)⁴⁷. This suggests that the identified Kac site on gut microbial PCK might be involved in accelerating the metabolic flow of PEP to oxaloacetate and thereby succinate/propionate (Fig. 3b). Taken together, these findings suggest that gut microbial Kac might be an important mechanism regulating the SCFA metabolism and influences the complex host-microbiome interactions in diseases.” (Discussion; Pages 12-13 Lines 367-408)

Minor

The authors need to make available in supplementary material the list of identified peptides with MS relevant associated statistics in a way that can facilitate their further analysis.

Reply:

We thank the reviewer for this comment and have added a Supplementary data file (Supplementary Data 1) in the manuscript, which listed all identified peptides in both acetylotomic and metaproteomic aliquots, with their modification status, charge state, PSM score, PEP, etc.

The motif analysis of acetylated peptides could be relegated to supplementary information or shown briefly as an additional panel in Figure 1.

Reply:

According to the reviewer’s suggestion, we have relegated Figure 2 to the supplementary figure and added two panels (f and e) to Fig. 1 to briefly demonstrate the motif of microbiome Kac sites.

The IBD results serve as an interesting proof-of-principle but the sample sizes are not large enough to identify variations that may still be meaningful. This should be addressed in the discussion. Figure 6 could as well be moved to supplementary and significant differences indicated in Figure 5.

Reply:

We thank the reviewer for this comment. Accordingly, we have clearly indicated this limitation of the current sample cohort in Discussion section and tuned down the statement on the findings comparing CD and control microbiomes in the revised manuscript in Page 14 Lines 440-442, as follows:

“The current study was limited by the number of CD patients, however, the findings provide valuable information for designing further studies to understand the functionality of the microbiome in CD.”

As per your suggestion, we have moved Figure 6 to supplementary information and significant differences have also been indicated in Figure 5 (now as Fig. 6 in the revised manuscript) with star (*) in the middle panel.

Other modifications made by the authors:

(1) We have reformatted the manuscript according to the submission checklist of *Nature Communications*. For example, we have changed “Figure” to “Fig.”, and “Figure S” to “Supplementary Fig.”. We have also re-worded the Abstract, subheadings as well as some main text of the manuscript to meet the word length limit required by the journal.

(2) We have added two contributing authors, namely Yidai Yang and Jean-François Couture, who performed the protein structural analyses in response to reviewers’ comments. Accordingly, the author contribution section was also updated.

Reviewers' Comments:

Reviewer #1:

Remarks to the Author:

The authors have addressed the critiques raised.

Reviewer #2:

Remarks to the Author:

In general, the authors have done a careful and diligent job in considering and responding to the various review comments. This included some new work to address concerns, along with very detailed comments of new text/figures added. While not perfect, the revisions reflect most of the major comments of the three reviewers - as such, this manuscript is greatly improved in terms of both content and clarity.

I have only two remaining concerns:

1. P.2, top - the authors' statements of CD in the Abstract are still too over-reaching. In response to reviewer comments, they have now toned down the tenor in the manuscript, but this abstract needs to be revised to soften the results of the relatively few CD samples, which bring up some issues of general conclusions.

2. p. 10, lines 295-302 - the addition of the temporal variability (or lack thereof) of the protein acetylation is good, but this occurs too late in the manuscript, and in fact too late even in the CD discussion. The reader will want to know this information far earlier in the text - the authors should see how they can either move this section or allude to the key message much earlier. This would give the reader some assurance that the acetylation variability is not stochastic.

Reviewer #3:

Remarks to the Author:

The authors have addressed the concerns raised. They have attempted to showcase the value of this resource by better describing which proteins are being modified and providing at least one example of structural mapping. This could have been extended but it already gives others an idea that there will be many examples of novel regulatory acetylation sites potentially found in these data. The authors also improved other aspects of the paper and are more cautious about the messages that can be derived from a relative small cohort size. I have no further concerns.

Point-by-point Response to Referees' Comments

REVIEWERS' COMMENTS:

Reviewer #1 (Remarks to the Author):

The authors have addressed the critiques raised.

Reply:

We thank the reviewer for the endorsement.

Reviewer #2 (Remarks to the Author):

In general, the authors have done a careful and diligent job in considering and responding to the various review comments. This included some new work to address concerns, along with very detailed comments of new text/figures added. While not perfect, the revisions reflect most of the major comments of the three reviewers - as such, this manuscript is greatly improved in terms of both content and clarity.

I have only two remaining concerns:

1. P.2, top - the authors' statements of CD in the Abstract are still too over-reaching. In response to reviewer comments, they have now toned down the tenor in the manuscript, but this abstract needs to be revised to soften the results of the relatively few CD samples, which bring up some issues of general conclusions.

Reply:

According to the reviewer's suggestion, we have removed the following sentence from the Abstract in the revised manuscript.

“Most decreased Kac sites in Crohn’s disease were from Firmicutes, while most increased sites were from Bacteroidetes.”

2. p. 10, lines 295-302 - the addition of the temporal variability (or lack thereof) of the protein acetylation is good, but this occurs too late in the manuscript, and in fact too late even in the CD discussion. The reader will want to know this information far earlier in the text - the authors should see how they can either move this section or allude to the key message much earlier. This would give the reader some assurance that the acetylation variability is not stochastic.

Reply:

We thank the reviewer for this valuable comment. Accordingly, we have moved the results on temporal variability into the beginning of Results Section 5 in the revised manuscript (Pages 8-9 Lines 246-259).

Reviewer #3 (Remarks to the Author):

The authors have addressed the concerns raised. They have attempted to showcase the value of this resource by better describing which proteins are being modified and providing at least one example of structural mapping. This could have been extended but it already gives others an idea that there will be many examples of novel regulatory acetylation sites potentially found in these data. The authors also improved other aspects of the paper and are more cautious about the messages that can be derived from a relative small cohort size. I have not further concerns.

Reply:

We thank the reviewer for his/her support and endorsement.